# LLM-Augmented Retrieval: Enhancing Retrieval Models Through Language Models and Doc-Level Embedding

## Abstract

Recent advancements in embedding-based retrieval, also known as dense retrieval, have shown state of the art results and demonstrated superior performance over traditional sparse or bag-of-words-based methodologies. This paper presents a model-agnostic document-level embedding framework enhanced by large language model (LLM) augmentation. The implementation of this LLM-augmented retrieval framework has significantly enhanced the efficacy of prevalent retriever models, including Bi-encoders (Contriever, DRAGON) and late-interaction models (ColBERTv2). Consequently, this approach has achieved state-of-the-art results on benchmark datasets such as LoTTE and BEIR, underscoring its potential to refine information retrieval processes.

## 1 Introduction

In the realm of information retrieval (IR), the quest for more precise and efficient methods to retrieve relevant information from a vast repository has been ongoing. Traditional IR systems predominantly relied on sparse retrieval techniques, such as the bag-of-words model(HaCohen-Kerner et al., 2020; Robertson et al., 1995; Zhang et al., 2010), which often fall short in capturing the semantic richness of the query and the documents due to their reliance on exact keyword matches. This limitation has paved the way for the emergence of embedding-based retrieval (Huang et al., 2020), also known as dense retrieval, which promises enhanced retrieval performance by leveraging deep learning models to understand and represent the semantic content of texts.

Embedding-based retrieval systems operate by transforming text into dense vector spaces where semantically similar texts are mapped close to each other. This transformation is typically achieved through the use of neural networks, particularly those pre-trained on large corpora(Chang et al., 2020), enabling the capture of deep semantic relationships that are not readily apparent through keyword matching alone. The vectors, or embeddings, generated by these models facilitate a more nuanced and context-aware retrieval process.

The Bi-encoder architecture (Cer et al., 2018; Karpukhin et al., 2020), commonly utilized in dense retrieval, comprises two encoders, often transformer models (Vaswani et al., 2017), that generate vector representations for user queries and documents or passages. These encoders may be shared or distinct. The relevance of documents to queries is determined by computing the similarity between these vectors, typically using dot product or cosine similarity. Conversely, Cross-encoders (Nogueira & Cho, 2019) integrate inputs early, enabling complex interactions between queries and documents. They concatenate the query and document to form a joint embedding vector, which is then used to assess document relevance in retrieval tasks. Cross-encoders generally surpass Bi-encoders in tasks requiring detailed interaction analysis.

Late-interaction models, such as ColBERT (Khattab & Zaharia, 2020), ColBERTv2 (Santhanam et al., 2021) or SPALDE++ (Formal et al., 2022), are model architectures that hybrids cross-encoder models and Bi-encoder models. Queries and documents are independently encoded into token-level vector representations. So in some sense, this is a bag of embedding vectors model. The interaction between these representations, which constitutes the "late interaction", involves computing the cosine similarity or dot product scores over the token-level vector embedding.

All model architectures necessitate informative embeddings of user queries and target documents. Essentially, the quality and quantity of textual information govern the accuracy and recall of the retrieved contexts when the model parameters are fixed. Query rewriting (Gottlob et al., 2014; He et al., 2016; Singh & Sharan, 2017; Xiong & Callan, 2015) is an effective method for enhancing query information from the user's side. Conversely, we hypothesize that enriching document embeddings can also improve text retrieval quality. Historically, scalable methods for augmenting document-related information were elusive, but the advent of large language models (LLMs) offers a solution.

Our contributions are threefold: 1) We introduce LLM-augmented retrieval, a model-agnostic framework 1 that enhances the contextual information in the vector embeddings of documents, thereby improving the performance of existing retrievers; 2) We propose a document-level embedding approach that integrates the pre-existing and newly-augmented contextual information; 3) We validate this framework across various models and extensive datasets, achieving state-of-the-art performance improvements over original models.

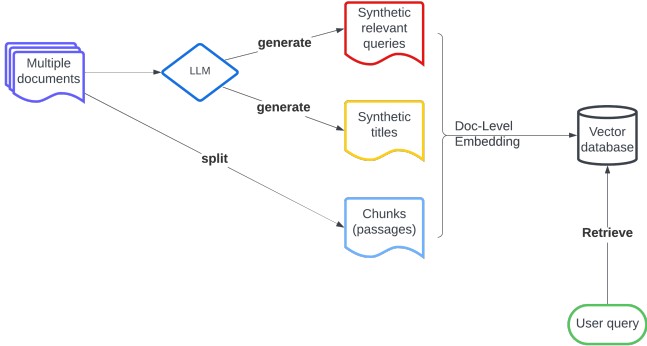

Figure 1: Overall view on LLM-augmented retrieval framework. Synthetic relevant queries and synthetic titles are generated from LLM and then assembled into doc-level embedding together with chunks (passages) split from the original document. The final retrieval is based on the similarity between user query and the doc-level embedding.

## 2 RELATED WORK

### 2.1 EMBEDDING-BASED RETRIEVAL

Recent advancements in the field of information retrieval have seen the integration of neural network architectures to compute text embeddings, which have shown to outperform the traditional sparse bag-of-words models in terms of effectiveness (Dai & Callan, 2019; Luan et al., 2021). Expanding on this foundation, Liu & Croft (2002) and Bendersky & Kurland (2008) have explored paragraph-based and window-based methods to delineate passages in information retrieval, respectively. Within the neural network domain, Fan et al. (2018) illustrated that aggregating representations to assess passage-level relevance yields promising results, particularly with pre-BERT models. Furthermore, Li et al. (2023a) introduced the technique of max-pooling to evaluate passage relevance. Our methodology draws upon similar principles to these preceding studies, aiming to further refine, aggregate and enhance the information from the the documents for embedding-based retrieval, through both max-pooling and average methods.

### 2.2 DATA AUGMENTATION AND PSEUDO QUERIES GENERATION

Data augmentation is a widely used technique in information retrieval training. Contrastive Learning (Izacard et al., 2021) has introduced techniques such as inverse cloze tasks, independent cropping, and random word deletion, replacement, or masking to enrich the diversity of training data. In

training the DRAGON model, Lin et al. (2023) studied query augmentation using query generation models and label augmentation methods with diverse supervision.

Pre-generated pseudo queries have been shown to be effective in improving retrieval performance. Previous works have calculated the similarity between pseudo-queries and user-queries using BM25 or BERT models to determine the final relevance score of the query to document through relevance score fusion (Chen et al., 2021; Wen et al., 2023). An alternative method for generating pseudo queries involves generating pseudo query embeddings through K-means clustering algorithms (Tang et al., 2021) or some fine-tuned models (Li et al., 2023b). Large pre-trained language models have demonstrated their ability to generate high-quality text data (Anaby-Tavor et al., 2020; Kumar et al., 2020; Meng et al., 2022; Schick & Schütze, 2021; Papanikolaou & Pierleoni, 2020; Yang et al., 2020). Some previous works have leveraged the generation capabilities of language models to create synthetic training data for retriever models fine-tuning (Bonifacio et al., 2022; Jeronymo et al., 2023; Nogueira et al., 2019; Wang et al., 2023). In our research, we employ large language models to generate pseudo queries similarly; however, these synthetic queries are utilized not during the training phase but at the inference stage of the retrieval system, specifically pre-calculated for the construction of the retrieval index. Our approach is training-free, requiring no fine-tuning, and leverages the foundational knowledge of LLMs for query generation, as well as the foundational knowledge of retrievers for calculating similarity scores. By eliminating the need for training, we can minimize costs and ensure that the method generalizes effectively across various scenarios.

# 3 LLM-AUGMENTED RETRIEVAL

This section introduces the components of the Large Language Model (LLM)-augmented retrieval framework and discusses its adaptability to various retriever model architectures. We propose the implementation of document-level embeddings for Bi-encoders and late-interaction encoders within this framework. The application of these adaptations is demonstrated to enhance the quality of end-to-end retrieval effectively.

## 3.1 LLM-AUGMENTED RETRIEVAL FRAMEWORK

### 3.1.1 SYNTHETIC RELEVANT QUERIES

The concept of synthetic relevant queries is inspired by established web search methodologies, as documented in several studies (Chuklin et al., 2022; Guo et al., 2009a;b; Xue et al., 2004). To elucidate this idea, consider the query "MIT". Without contextual knowledge, the equivalence between "MIT" and "Massachusetts Institute of Technology" may not be immediately apparent. In web search contexts, the frequent selection (clicks) of the Massachusetts Institute of Technology's homepage in response to the query "MIT" suggests a strong association between the two. This inference is drawn from observed user interactions, which are often unavailable in contextual retrieval scenarios. In such cases where direct click data and frequent-clicked relevant queries are absent, large language models have demonstrated proficiency in generating synthetic queries that can serve as surrogate indicators of user interest (Anaby-Tavor et al., 2020; Kumar et al., 2020; Meng et al., 2022; Papanikolaou & Pierleoni, 2020; Schick & Schütze, 2021; Yang et al., 2020). These synthetic queries effectively mimic "frequent-clicked relevant queries", guiding the alignment of user queries with pertinent documents.

A critical aspect to consider is the traditional reliance on similarity metrics to determine relevance in retrieval tasks (Jones & Furnas, 1987). These metrics, typically the dot product or cosine similarity of encoded vectors, may not always capture the semantic nuances essential for relevance. For instance, the queries "Who is the first president of the United States?" and "Who became the first president of America?" might yield high similarity scores but diverge in semantic relevance. The desired document, such as a biography of George Washington, might not score as highly against these queries. However, if synthetic queries generated from Washington's biography include "Who became the first president of America?", it becomes possible to bridge the semantic gap. The synthetic query not only reflects the document's content from various perspectives but also enhances the matching process with relevant user queries, as illustrated in Figure 2a. This approach underscores the utility of synthetic queries in capturing and conveying the semantic essence of documents, thereby improving the alignment with user-intended search outcomes.

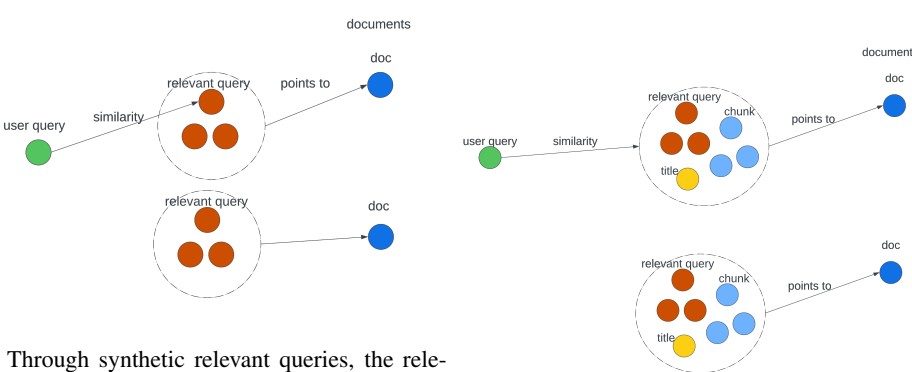

(a) Through synthetic relevant queries, the relevance relationship is not solely expressed by the similarity now but also expressed by the augmentation steps of the large language models

(b) The graphic representation of "relevance" in doc-level embedding

Figure 2: Graphic representation of synthetic queries, titles, passage chunks in doc-level embedding

### 3.1.2 TITLE

The title of a document is pivotal in determining its relevance and utility in response to a user's search query. As the primary element encountered by users in search results, the title significantly influences their decision-making process regarding which links to pursue. An effectively formulated title furnishes essential context and keywords, enabling users to swiftly ascertain the content and objective of a document.

In instances where the original document possesses a title, it can be directly utilized to enhance search relevance. Conversely, for documents lacking a title, the deployment of large language models becomes instrumental. These models are capable of generating synthetic titles that encapsulate the essence and main themes of the document. This capability not only aids in accurately representing the document's content but also in aligning it more closely with the informational needs expressed in user queries. Thus, whether derived directly or synthesized through advanced modeling, titles play a crucial role in optimizing the search and discovery process.

### 3.1.3 DOCUMENT CHUNKS

Document chunking (Chen et al., 2023; Finardi et al., 2024; Lewis et al., 2020) is a methodological approach that involves segmenting a large document or text into smaller, more digestible units referred to as "chunks" or "passages." This process typically groups together related segments of information to facilitate more manageable analysis and processing. The necessity for chunking arises primarily due to the constraints imposed by the context window of retrieval models, which limits the maximum length of model input.

In practice, a lengthy document is divided into several chunks, each containing a number of tokens that do not exceed the model's context window limit. It is important to note that these chunks are derived directly from the original documents without augmentation from large language models (LLMs).

The determination of an optimal chunk size for Bi-encoders varies across different retrieval models. Conversely, token-level late-interaction models like ColBERT and ColBERTv2, which calculate similarity scores at the token level, do not require chunking of the original documents unless the context window limit is exceeded. This distinction underscores the model-specific considerations that must be taken into account when implementing chunking strategies in information retrieval systems.

## 3.2 DOC-LEVEL EMBEDDING

This section introduces the concept of document-level embedding for information retrieval and illustrates its adaptability across different retriever model structures, including Bi-encoders and token-level late-interaction models.

**Definition 3.1.** Document Fields: Synthetic queries, titles, and chunks constitute the **fields** of a document.

For clarity, we refer to these information sources—synthetic queries, titles, and chunks—as the fields of a document. These fields represent the semantics of the original document from various perspectives and are integrated into the document-level embedding (see Figure 2b). This embedding is static, allowing it to be pre-computed and cached for efficient retrieval. Indexes of these embeddings can be pre-built to expedite the retrieval process, with each embedding linking back to the original document.

### 3.2.1 FOR BI-ENCODERS

Bi-encoders are typically structured as "Two-Tower" models. In this configuration, separate encoders process a query and a document to generate respective embedding vectors. These vectors are then utilized to calculate similarity scores through dot products or cosine similarity measures. To enhance the document embeddings by incorporating synthetic queries and titles, we propose the following similarity computation:

**Definition 3.2.** Similarity score for query-document pairs in Bi-encoders:

$$sim(q, d) = \max_i s(q, c_i) + s(q, d) \tag{1}$$

$$\text{where } s(q,d) = s(q, \frac{w_c}{m} \sum_i^m c_i + \frac{w_{q*}}{n} \sum_j^n q_j^* + w_{t*} t^*) \tag{2}$$

The term $\max_i s(q, c_i)$ computes the traditional maximum similarity score across query-chunk embedding pairs, where $s$ denotes the similarity function, $q$ represents the search query's embedding vector, and $c_i$ is the embedding vector for the $i$-th document chunk. This approach is prevalent in modern embedding-based retrieval systems, focusing on the similarity between a query and the most relevant document chunk. The second term $s(q, d)$ introduces a novel aspect by incorporating additional augmented information at document level. Here, $c$ are the chunk embedding vectors mentioned above, $q*$ are the embedding vectors synthetic relevant queries, $t^*$ is the title embedding vector, while $w_c$, $w_q^*$, $w_{t*}$ are the corresponding document field weights. Arora et al. (2017) also suggests averaging these vectors to represent the entire document, as an approach we adapt for both chunk and synthetic query fields. This method has proven effective in our experiments, though more sophisticated techniques could be explored in future work.

Given that the similarity function is linear[1], the equation can be transformed to:

$$sim(q, d) = \max_i s(q, c_i + \frac{w_c}{m} \sum_i^m c_i + \frac{w_{q*}}{n} \sum_j^n q_j^* + w_{t*} t^*) \tag{3}$$

This simplification allows us to treat $c_i + \frac{w_c}{m} \sum_i^m c_i + \frac{w_{q*}}{n} \sum_j^n q_j^* + w_{t*} t^*$ as the composite embedding vector for each document chunk $c_i$, enabling the use of algorithms like approximate nearest neighbors (Indyk & Motwani, 1998) for efficient document retrieval.

### 3.2.2 FOR TOKEN-LEVEL LATE-INTERACTION MODELS

Late-interaction models such as ColBERT and ColBERTv2 diverge from traditional approaches by utilizing token-level embeddings rather than a single embedding vector for both the query and each document. In these models, embeddings for all tokens are retained and contribute to the computation of the similarity score between the query and the document.

---

[1] Both dot product and cosine similarity are linear when embedding vectors are normalized to unit length.

**Definition 3.3.** Similarity score for query-document pairs in token-Level late-interaction models:

$$sim(q,d) = \sum_i \max_j s(q_i, T_j) \tag{4}$$

where $q_i$ and $T_j$ represent the token-level embedding vectors for the input query and the document, respectively. For each token in the query, the model identifies the most similar corresponding token in the document, and the similarity score for these token pairs is calculated. The overall similarity between the query and the document is then determined by summing these scores across all query tokens. Therefore, it becomes feasible to augment the original document passages with synthetic queries and titles to calculate the query-document similarity scores. Subsequently, if the total number of tokens exceeds the context window limit, a decision can be made regarding the chunking of the concatenated documents. This method allows for a more granular and potentially more accurate matching process between queries and documents.

## 4 EXPERIMENTS

### 4.1 DATASETS

**BEIR Data**
The BEIR (Benchmark for Evaluating Information Retrieval) dataset (Thakur et al., 2021) serves as a comprehensive benchmark for assessing various information retrieval (IR) models, particularly in out-of-domain scenarios. Designed to overcome the limitations of previous datasets, BEIR offers a diverse and extensive collection of queries and passages across a broad range of topics. This diversity enables a more thorough and robust evaluation of IR models.

**LoTTE Data**
The LoTTE dataset (Santhanam et al., 2021) is specifically crafted for Long-Tail Topic-stratified Evaluation, focusing on natural user queries linked to long-tail topics that are often underrepresented in entity-centric knowledge bases like Wikipedia. Comprising 10 distinct test sets, each containing 500 to 2,000 queries and 100,000 to 2,000,000 passages, these sets are categorized by topic. Each test set is paired with a validation set that includes related but disjoint queries and passages. For this experiment, only the test split is utilized for evaluation purposes.

### 4.2 MODELS

**Contriever**
The Contriever model employs the Roberta-base (Liu et al., 2019) architecture, trained on Wiki passages (Karpukhin et al., 2020) and CC100 (Conneau et al., 2019) data through contrastive learning. It features 125 million parameters, a context window of 512 tokens, 12 layers, 768 hidden dimensions, and 12 attention heads. In this model, a single Roberta-base model serves as both the query encoder and context encoder, following a shared "Two Tower" Bi-encoder architecture.

**DRAGON**
Similarly, the DRAGON model utilizes the Roberta-base architecture. However, unlike Contriever, DRAGON employs separate Roberta-base models for the query encoder and context encoder. This model's checkpoint was trained and released publicly by the author.

**ColBERTv2**
For ColBERTv2, the bert-base-uncased model architecture is adopted, consistent with the default settings in the original paper. This model comprises 110 million parameters and a context window of 256 tokens, with 12 layers, 768 hidden dimensions, and 12 attention heads. The checkpoint for ColBERTv2 was trained on the MSMARCO dataset (Nguyen et al., 2016) and provided by the author.

### 4.3 IMPLEMENTATION DETAILS

We choose open source Llama-70B (Dubey et al., 2024; Touvron et al., 2023a;b) for synthetic query generation and title generation. The prompt templates used for generating synthetic queries and titles are in Table 9 and 10.

For Bi-encoders, we implemented the doc-level embedding as above mentioned with chunk_size=64 and chose $w_{q*}$=1.0, $w_{t*}$=0.5, $w_c$=0.1 for the Contriever model and $w_{q*}$=0.6, $w_{t*}$=0.3, $w_c$=0.3 for the DRAGON model. We used the dev set of BEIR-ArguAna to choose all the hyperparameters and fix the hyperparameters across all the evaluation sets. The hyperparameters seem to generalize really well. For ColBERTv2, as mentioned previously, we concatenate the title with all the synthetic queries for each document and make it an additional "passage" of the original document. Thus there's no field weights hyper-parameters in these experiments. There could be other better assembling methods for composing the doc-level embedding under a late-interaction model architecture. We set index_bits=8 when building the ColBERT index.

## 4.4 RESULTS

The results for the three models on the LoTTE and BEIR datasets are presented in Tables 1, 2, and 3. It is evident that the integration of large language model (LLM) augmented retrieval and document-level embeddings significantly enhances the Recall@3 (R@3) and Recall@10 (R@10) metrics for Bi-encoder models (Contriever and DRAGON). For token-level late-interaction models such as ColBERTv2, there is a noticeable improvement in performance on the LoTTE and BEIR datasets, though the magnitude of enhancement is less pronounced compared to the Bi-encoders. This discrepancy is hypothesized to stem from the higher baseline performance of token-level late-interaction models relative to Bi-encoders.

Furthermore, the performance of the LLM-augmented Contriever surpasses that of the standard DRAGON across most datasets. In a similar vein, the LLM-augmented DRAGON outperforms the standard ColBERTv2 on specific datasets such as BEIR-ArguAna, BEIR-SciDocs, and BEIR-CQADupstack-English, and significantly narrows the performance gap on other datasets. This occurs despite ColBERTv2's more intricate late-interaction architecture compared to DRAGON.

| LoTTE - Search | | | | | | |
|---|---|---|---|---|---|---|
| Model | Recall | Lifestyle Search | Recreation Search | Science Search | Technology Search | Writing Search |
| Contriever | R@3 | 0.3358 | 0.1948 | 0.1005 | 0.1242 | 0.2745 |
| | R@10 | 0.4690 | 0.2857 | 0.1637 | 0.1896 | 0.3950 |
| Contriever* | R@3 | **0.6021** | **0.4610** | **0.2901** | **0.3557** | **0.5724** |
| | R@10 | **0.7821** | **0.6320** | **0.4684** | **0.5017** | **0.6919** |

| LoTTE - Forum | | | | | | |
|---|---|---|---|---|---|---|
| Model | Recall | Lifestyle Forum | Recreation Forum | Science Forum | Technology Forum | Writing Forum |
| Contriever | R@3 | 0.4366 | 0.3486 | 0.1046 | 0.1826 | 0.3950 |
| | R@10 | 0.6149 | 0.4895 | 0.1706 | 0.3174 | 0.5390 |
| Contriever* | R@3 | **0.6244** | **0.5455** | **0.2395** | **0.3663** | **0.5970** |
| | R@10 | **0.7622** | **0.6948** | **0.3570** | **0.5494** | **0.7365** |

| BEIR | | | | | | | |
|---|---|---|---|---|---|---|---|
| Model | Recall | ArguAna | FIQA | Quora | SciDocs | SciFact | CQAD English | CQAD Physics |
| Contriever | R@3 | **0.2589** | 0.1895 | 0.8654 | 0.1580 | 0.5410 | 0.2261 | 0.1723 |
| | R@10 | 0.5206 | 0.2993 | 0.9463 | 0.2950 | 0.6934 | 0.3089 | 0.2551 |
| Contriever* | R@3 | 0.2468 | **0.3690** | **0.8687** | **0.2440** | **0.5996** | **0.3822** | **0.3417** |
| | R@10 | **0.5825** | **0.5174** | **0.9517** | **0.4030** | **0.7259** | **0.5025** | **0.4658** |

Table 1: Results on Contriever: The performance of LLM-augmented Contriever has greatly exceeded the vanilla Contriever on both LoTTE and BEIR dataset, and even exceeds the performance of the vanilla DRAGON in most datasets. Contriever* means base model plus the doc-level embedding ($w_{q*}$=1.0, $w_{t*}$=0.5, $w_c$=0.1).

**LoTTE - Search**

| Model | Recall | Lifestyle Search | Recreation Search | Science Search | Technology Search | Writing Search |
|---|---|---|---|---|---|---|
| DRAGON | R@3 | 0.5598 | 0.4253 | 0.2601 | 0.3591 | 0.5798 |
|  | R@10 | 0.7035 | 0.5325 | 0.3938 | 0.5101 | 0.7311 |
| DRAGON* | R@3 | **0.7625** | **0.6472** | **0.4498** | **0.5285** | **0.7031** |
|  | R@10 | **0.8911** | **0.7944** | **0.6062** | **0.7097** | **0.8170** |

**LoTTE - Forum**

| Model | Recall | Lifestyle Forum | Recreation Forum | Science Forum | Technology Forum | Writing Forum |
|---|---|---|---|---|---|---|
| DRAGON | R@3 | 0.5270 | 0.4560 | 0.2578 | 0.2854 | 0.5300 |
|  | R@10 | 0.6798 | 0.5949 | 0.3704 | 0.4232 | 0.6675 |
| DRAGON* | R@3 | **0.6883** | **0.6079** | **0.3099** | **0.4192** | **0.6520** |
|  | R@10 | **0.8172** | **0.7468** | **0.4427** | **0.6038** | **0.7725** |

**BEIR Dataset**

| Model | Recall | ArguAna | FIQA | Quora | SciDocs | SciFact | CQAD English | CQAD Physics |
|---|---|---|---|---|---|---|---|---|
| DRAGON | R@3 | 0.1408 | 0.3327 | 0.8465 | 0.1800 | 0.4743 | 0.2605 | 0.1877 |
|  | R@10 | 0.4040 | 0.4514 | 0.9419 | 0.3260 | 0.5996 | 0.3599 | 0.2916 |
| DRAGON* | R@3 | **0.3663** | **0.4255** | **0.8638** | **0.3040** | **0.6610** | **0.4618** | **0.3936** |
|  | R@10 | **0.6764** | **0.5635** | **0.9527** | **0.4800** | **0.7710** | **0.5662** | **0.5342** |

Table 2: Results on DRAGON: The performance of LLM-augmented DRAGON has greatly exceeded the vanilla DRAGON on both LoTTE and BEIR dataset, and even exceeds vanilla ColBERTv2 on BEIR-ArguAna, BEIR-SciDocs and BEIR-CQADupstack-English datasets, as well as greatly reduces the performance gap in the remaining datasets. DRAGON* means base model plus the doc-level embedding ($w_{q^*}$=0.6, $w_{t^*}$=0.3, $w_c$=0.3).

**LoTTE - Search**

| Model | Recall | Lifestyle Search | Recreation Search | Science Search | Technology Search | Writing Search |
|---|---|---|---|---|---|---|
| ColBERTv2 | R@3 | 0.7927 | 0.6677 | 0.5073 | 0.5940 | 0.7423 |
|  | R@10 | 0.8911 | 0.7868 | 0.6613 | 0.7315 | 0.8366 |
| ColBERTv2* | R@3 | **0.8003** | **0.7100** | **0.5024** | **0.5956** | **0.7544** |
|  | R@10 | **0.9107** | **0.8268** | **0.6726** | **0.7383** | **0.8571** |

**LoTTE - Forum**

| Model | Recall | Lifestyle Forum | Recreation Forum | Science Forum | Technology Forum | Writing Forum |
|---|---|---|---|---|---|---|
| ColBERTv2 | R@3 | 0.6988 | 0.6344 | 0.3932 | 0.4496 | 0.6960 |
|  | R@10 | 0.8087 | 0.7498 | 0.5285 | 0.6292 | 0.8050 |
| ColBERTv2* | R@3 | **0.7308** | **0.6753** | **0.4026** | **0.4626** | **0.7145** |
|  | R@10 | **0.8447** | **0.7862** | **0.5558** | **0.6517** | **0.8260** |

**BEIR Dataset**

| Model | Recall | ArguAna | FIQA | Quora | SciDocs | SciFact | CQAD English | CQAD Physics |
|---|---|---|---|---|---|---|---|---|
| ColBERTv2 | R@3 | 0.3542 | 0.4469 | 0.9048 | 0.2990 | 0.6691 | 0.4484 | 0.4052 |
|  | R@10 | 0.6287 | 0.5787 | 0.9643 | 0.4780 | 0.7755 | 0.5369 | 0.5380 |
| ColBERTv2* | R@3 | **0.3592** | **0.4666** | **0.9067** | **0.3000** | **0.6862** | **0.4822** | **0.4196** |
|  | R@10 | **0.6344** | **0.6018** | **0.9663** | **0.4850** | **0.7917** | **0.5694** | **0.5611** |

Table 3: Results on ColBERTv2: The performance of LLM-augmented ColBERTv2 has greatly exceeded the performance of vanilla ColBERTv2 on both LoTTE and BEIR dataset. ColBERTv2* means base model plus the doc-level embedding.

## 4.5 AUGMENTATION ANALYSIS

Table 4 gives an overview on the number of documents per dataset ($N_D$, in thousands), the number of total tokens in documents ($N_{T_D}$, in thousands), the average number of tokens per document ($N_{T_D}/N_D$), the number of synthetic queries generated ($N_{q^*}$, in thousands), the total number of total synthetic query tokens generated ($N_{T_{q^*}}$, in thousands), the average number of synthetic query per document ($N_{q^*}/N_D$), the average number of synthetic query tokens per document ($N_{T_{q^*}}/N_D$) and the average number of synthetic query tokens per synthetic query ($N_{T_{q^*}}/N_{q^*}$). On average 6 synthetic relevant queries are generated per document and the token count in the generated synthetic queries is comparable to the token count in the original documents. The average ratio of synthetic query tokens to original document tokens ($N_{T_{q^*}}/N_{T_D}$) is 57% and this ratio decreases to 51% when the Quora dataset is excluded. While the number of generated tokens is comparable to that of the original tokens, our method involves only a single decoding (generation) and encoding (retrieval index construction) step throughout the entire procedure. Furthermore, our method does not require any training, rendering it costing less than traditional query augmentation techniques that rely on augmented queries solely for retriever model training. Additionally, the inference speed remains unaffected, as the retrieval index is pre-constructed using the augmented tokens.

We also compute the query match ratio, denoted as Match($q^*$), which is defined as the ratio of the number of intersections between search queries and synthetic relevant queries to the total number of search queries. This metric is reported in Table 4. It is observed that most Match($q^*$) values are zero, with the exceptions being the Quora and FIQA datasets.

| Dataset | Subset | Original Documents | | | Generated Synthetic Relevant Queries | | | | | |
|---|---|---|---|---|---|---|---|---|---|---|
| | | $N_D$ (in K) | $N_{T_D}$ (in K) | $N_{T_D}/N_D$ | $N_{q^*}$ (in K) | $N_{T_{q^*}}$ (in K) | $N_{q^*}/N_D$ | $N_{T_{q^*}}/N_D$ | $N_{T_{q^*}}/N_{q^*}$ | Match($q^*$) % |
| BEIR | ArguAna | 9 | 1,782 | 205 | 46 | 684 | 5 | 79 | 15 | 0 |
| | FIQA | 58 | 9,470 | 164 | 305 | 4,360 | 5 | 76 | 14 | 1.0 |
| | Quora | 523 | 8,404 | 16 | 3,123 | 40,947 | 6 | 78 | 13 | 6.2 |
| | SciDocs | 25 | 5,365 | 212 | 160 | 2,580 | 6 | 102 | 16 | 0 |
| | SciFact | 5 | 1,548 | 299 | 32 | 618 | 6 | 119 | 19 | 0 |
| | CQAD English | 40 | 4,251 | 106 | 179 | 2,987 | 4 | 74 | 17 | 0 |
| | CQAD Physics | 38 | 6,992 | 182 | 184 | 3,232 | 5 | 84 | 18 | 0 |
| LoTTE | Lifestyle | 119 | 21,639 | 181 | 664 | 9,866 | 6 | 83 | 15 | 0 |
| | Recreation | 167 | 26,988 | 162 | 902 | 13,215 | 5 | 79 | 15 | 0 |
| | Science | 1,694 | 400,544 | 236 | 8,461 | 159,901 | 5 | 94 | 19 | 0 |
| | Technology | 662 | 117,940 | 178 | 7,031 | 105,610 | 11 | 159 | 15 | 0 |
| | Writing | 200 | 29,031 | 145 | 1,027 | 15,364 | 5 | 77 | 15 | 0 |

Table 4: Statistics on original document information and augmented document information for each dataset

## 4.6 ABLATION STUDIES

### 4.6.1 STUDY ON EFFECT OF DIFFERENT LLMS USED FOR GENERATION

In this section, we also compare the performance difference between different LLMs (Llama2-7b, Llama2-70b, Llama3-8b and Llama3-70) for synthetic query generaiton and summarized the evaluation results on two BEIR datasets in Table 5. Table 11 further provides several high-quality examples of the generated synthetic queries from four selected documents. The patterns of queries generated by different LLMs and their corresponding recall performance show minimal variation.

| Model | Dataset | Metrics | Llama2-7b | Llama2-70b | Llama3-8b | Llama3-70b |
|---|---|---|---|---|---|---|
| Contriever* | ArguAna | R@3 | 0.2425 | 0.2468 | 0.2447 | 0.2596 |
| | | R@10 | 0.5583 | 0.5825 | 0.5939 | 0.6110 |
| | SciFact | R@3 | 0.5870 | 0.5996 | 0.5996 | 0.6231 |
| | | R@10 | 0.7106 | 0.7259 | 0.7196 | 0.7430 |
| Dragon* | ArguAna | R@3 | 0.4132 | 0.3663 | 0.4232 | 0.4289 |
| | | R@10 | 0.7269 | 0.6764 | 0.7496 | 0.7624 |
| | SciFact | R@3 | 0.6303 | 0.6610 | 0.6348 | 0.6528 |
| | | R@10 | 0.7520 | 0.7710 | 0.7538 | 0.7592 |

Table 5: Comparison on synthetic relevant queries generated by different models

### 4.6.2 STUDY ON THE EFFECT OF SYNTHETIC RELEVANT QUERIES AND TITLES

This section explores the impact of LLM-augmented document fields—query and title—on the retrieval quality of various retriever models. For Bi-encoders (Contriever and DRAGON), we manipulate the field weights of synthetic query, and title to examine their influence on performance metrics. Conversely, for the token-level late-interaction model (ColBERTv2), we isolate each field (chunk, query, or title) to assess its individual effect on end-to-end retrieval quality.

In the case of the Contriever model (Table 6), synthetic queries generally play a pivotal role in enhancing recall performance compared to the other fields. However, their relative importance diminishes in datasets such as BEIR-SciDocs and BEIR-Scifact. It appears that a weighted combination of multiple fields in document-level embeddings tends to yield superior performance in most scenarios, suggesting that these weights could be optimized as hyperparameters.

For the DRAGON model (Table 7), no consistent pattern emerges regarding which field most significantly influences document-level embedding. In the LoTTE dataset, the title field appears to be more influential. Similar to the Contriever model, integrating multiple document fields into a weighted sum generally improves performance. The observed differences between DRAGON and Contriever may be attributed to DRAGON's use of separate query and context encoders, as opposed to Contriever's shared encoders. This architectural distinction likely makes Contriever more adept at capturing similarity rather than relevance, thereby enhancing the impact of synthetic queries in transforming similarity into relevance.

Regarding ColBERTv2 (Table 8), cross the datasets, synthetic queries are found to be more crucial than titles for ColBERTv2, and combining all fields typically results in even better recall outcomes. It is important to note that there are no field weight hyperparameters for token-level late-interaction models.

## 5 CONCLUSION

This paper introduces a novel framework termed LLM-augmented retrieval, which substantially enhances the performance of existing retriever models by augmenting document embeddings with large language model (LLM) inputs. This framework incorporates document-level embeddings that encode contextual information derived from synthetic queries, titles, and chunks, and is adaptable to various retriever model architectures. The implementation of this approach has yielded state-of-the-art results across multiple models and datasets, affirming its efficacy in improving the quality of neural information retrieval. Future research could focus on further refinements to the LLM-augmented retrieval framework, such as incorporating more diverse contextual information into document-level embeddings, employing more sophisticated measures for similarity scoring, and developing more complex methods for integrating multiple chunks or queries into a single field embedding.

## 6 LIMITATIONS

This study encounters several limitations, notably the increased computational resources required for augmenting relevant queries and titles for the original documents. In some instances, the size of the augmented texts may approach or equal that of the original documents, which could pose a significant computational burden. This limitation may hinder the applicability of this approach in environments where computational resources are constrained.

Another potential limitation concerns the risk of hallucination in large language models, which can introduce inaccuracies into the augmented corpus relative to the original documents. Hallucination remains a persistent challenge in the field of large language model research and could compromise the integrity of the retrieval process.

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

# A  APPENDIX

## A.1  ABLATION STUDY

| LoTTE - Search | | | | | | |
|---|---|---|---|---|---|---|
| Model | Recall | Lifestyle Search | Recreation Search | Science Search | Technology Search | Writing Search |
| Contriever $w_{q*}$=1.0 | R@3 | **0.6967** | 0.4437 | 0.2901 | 0.3305 | 0.5472 |
| | R@10 | **0.7837** | 0.6115 | 0.4295 | 0.4883 | 0.6910 |
| Contriever $w_{t*}$=1.0 | R@3 | 0.4902 | 0.3789 | 0.1896 | 0.2668 | 0.4809 |
| | R@10 | 0.6641 | 0.5314 | 0.3241 | 0.4077 | 0.6153 |
| Contriever* | R@3 | 0.6021 | **0.4610** | **0.2901** | **0.3557** | **0.5724** |
| | R@10 | 0.7821 | **0.6320** | **0.4684** | **0.5017** | **0.6919** |

| LoTTE - Forum | | | | | | |
|---|---|---|---|---|---|---|
| Model | Recall | Lifestyle Forum | Recreation Forum | Science Forum | Technology Forum | Writing Forum |
| Contriever $w_{q*}$=1.0 | R@3 | 0.6194 | 0.5355 | 0.2335 | 0.3523 | 0.5860 |
| | R@10 | 0.7762 | 0.6863 | 0.3461 | 0.5180 | 0.7410 |
| Contriever $w_{t*}$=1.0 | R@3 | 0.5310 | 0.4975 | 0.2345 | 0.3468 | 0.5315 |
| | R@10 | 0.6958 | 0.6404 | 0.3421 | 0.5200 | 0.6725 |
| Contriever* | R@3 | **0.6244** | **0.5455** | **0.2395** | **0.3663** | **0.5970** |
| | R@10 | **0.7622** | **0.6948** | **0.3570** | **0.5494** | **0.7365** |

| BEIR Dataset | | | | | | | |
|---|---|---|---|---|---|---|---|
| Model | Recall | ArguAna | FIQA | Quora | SciDocs | SciFact | CQAD English | CQAD Physics |
| Contriever $w_{q*}$=1.0 | R@3 | 0.2347 | 0.3580 | 0.8622 | 0.2180 | 0.5888 | **0.3860** | 0.3330 |
| | R@10 | 0.5718 | 0.5045 | 0.8088 | 0.3720 | 0.7322 | 0.5013 | 0.4629 |
| Contriever $w_{t*}$=1.0 | R@3 | 0.2063 | 0.3180 | 0.7555 | **0.2600** | 0.5573 | 0.3338 | 0.2926 |
| | R@10 | 0.5192 | 0.4595 | 0.8791 | **0.4120** | 0.7051 | 0.4369 | 0.4100 |
| Contriever* | R@3 | **0.2468** | **0.3690** | **0.8687** | 0.2440 | 0.5996 | 0.3822 | **0.3417** |
| | R@10 | **0.5825** | **0.5174** | **0.9517** | 0.4030 | 0.7259 | **0.5025** | **0.4658** |

Table 6: Ablation study on doc-level embedding with Contriever. In most cases the ensemble of relevant queries, title and chunks gives the best results. Contriever* means base model plus the doc-level embedding (chunk:0.1, query:1.0, title:0.5).

**LoTTE - Search**

| Model | Recall | Lifestyle Search | Recreation Search | Science Search | Technology Search | Writing Search |
|-------|--------|------------------|-------------------|----------------|-------------------|----------------|
| DRAGON | R@3 | 0.7247 | 0.6071 | 0.3647 | 0.4866 | 0.6583 |
| $w_{q*}$=1.0 | R@10 | 0.8654 | 0.7478 | 0.5219 | 0.6812 | 0.7656 |
| DRAGON | R@3 | 0.7610 | 0.6472 | 0.4408 | **0.5436** | 0.6928 |
| $w_{t*}$=1.0 | R@10 | 0.8790 | 0.7879 | 0.5948 | 0.7064 | 0.8011 |
| DRAGON* | R@3 | **0.7625** | **0.6472** | **0.4498** | 0.5285 | **0.7031** |
| | R@10 | **0.8911** | **0.7944** | **0.6062** | **0.7097** | **0.8170** |

**LoTTE - Forum**

| Model | Recall | Lifestyle Forum | Recreation Forum | Science Forum | Technology Forum | Writing Forum |
|-------|--------|-----------------|------------------|---------------|------------------|---------------|
| DRAGON | R@3 | 0.6583 | 0.5839 | 0.2707 | 0.3892 | 0.6235 |
| $w_{q*}$=1.0 | R@10 | 0.8017 | 0.7108 | 0.3991 | 0.5704 | 0.7420 |
| DRAGON | R@3 | **0.6913** | **0.6294** | **0.3565** | **0.4616** | **0.6550** |
| $w_{t*}$=1.0 | R@10 | 0.8167 | 0.7458 | **0.4834** | **0.6477** | 0.7690 |
| DRAGON* | R@3 | 0.6883 | 0.6079 | 0.3099 | 0.4192 | 0.6520 |
| | R@10 | **0.8172** | **0.7468** | 0.4427 | 0.6038 | **0.7725** |

**BEIR Dataset**

| Model | Recall | ArguAna | FIQA | Quora | SciDocs | SciFact | CQAD English | CQAD Physics |
|-------|--------|---------|------|-------|---------|---------|--------------|--------------|
| DRAGON | R@3 | 0.3265 | 0.3875 | 0.8267 | 0.2820 | 0.6032 | 0.4318 | 0.3503 |
| $w_{q*}$=1.0 | R@10 | 0.6472 | 0.5220 | 0.9283 | 0.4470 | 0.7403 | 0.5344 | 0.4889 |
| DRAGON | R@3 | 0.3208 | **0.4310** | 0.8039 | 0.2940 | 0.6375 | 0.4516 | **0.4081** |
| $w_{t*}$=1.0 | R@10 | 0.6230 | **0.5692** | 0.9139 | 0.4770 | 0.7556 | 0.5567 | 0.5274 |
| DRAGON* | R@3 | 0.3663 | 0.4255 | **0.8638** | **0.3040** | **0.6610** | **0.4618** | 0.3936 |
| | R@10 | 0.6764 | 0.5635 | **0.9527** | **0.4800** | 0.7710 | **0.5662** | **0.5342** |

Table 7: Ablation study on doc-level embedding with DRAGON. In most cases the ensemble of relevant queries, title and chunks gives the best results. DRAGON* means base model plus the doc-level embedding (chunk:0.3, query:0.6, title:0.3).

## A.2 PROMPTS FOR GENERATING SYNTHETIC RELEVANT QUERIES AND TITLE

I will give you an article below. What are some search queries or questions that are relevant for this article or this article can answer?
Separate each query in a new line.
This is the article: {document}
Only provide the user queries without any additional text. Format every query as 'query:' followed by the question. Don't write empty queries.

Table 9: Prompt for generating relevant queries for documents

I will give you an article below. Create a title for the below article.
This is the article: {document}
Only provide the title without any additional text. Format the reply starting with 'title:' followed by the question. Don't write empty title.

Table 10: Prompt for generating titles for documents.

**LoTTE - Search**

| Model | Recall | Lifestyle Search | Recreation Search | Science Search | Technology Search | Writing Search |
|---|---|---|---|---|---|---|
| ColBERTv2 | R@3 | 0.7413 | 0.6580 | 0.4327 | 0.4530 | 0.7274 |
| $q^*$ only | R@10 | 0.8759 | 0.7727 | 0.5997 | 0.5419 | 0.8254 |
| ColBERTv2 | R@3 | 0.6218 | 0.5487 | 0.3695 | 0.4715 | 0.5780 |
| $t^*$ only | R@10 | 0.7458 | 0.6937 | 0.5024 | 0.6141 | 0.6853 |
| ColBERTv2* | R@3 | **0.8003** | **0.7100** | **0.5024** | **0.5956** | **0.7544** |
| | R@10 | **0.9107** | **0.8268** | **0.6726** | **0.7383** | **0.8571** |

**LoTTE - Forum**

| Model | Recall | Lifestyle Forum | Recreation Forum | Science Forum | Technology Forum | Writing Forum |
|---|---|---|---|---|---|---|
| ColBERTv2 | R@3 | 0.7088 | 0.6479 | 0.3634 | 0.3643 | 0.6835 |
| $q^*$ only | R@10 | 0.8222 | 0.7642 | 0.4948 | 0.5259 | 0.7890 |
| ColBERTv2 | R@3 | 0.6004 | 0.5210 | 0.3128 | 0.4336 | 0.5425 |
| $t^*$ only | R@10 | 0.7368 | 0.6479 | 0.4378 | 0.5968 | 0.6505 |
| ColBERTv2* | R@3 | **0.7308** | **0.6753** | **0.4026** | **0.4626** | **0.7145** |
| | R@10 | **0.8447** | **0.7862** | **0.5558** | **0.6517** | **0.8260** |

**BEIR Dataset**

| Model | Recall | ArguAna | FIQA | Quora | SciDocs | SciFact | CQAD English | CQAD Physics |
|---|---|---|---|---|---|---|---|---|
| ColBERTv2 | R@3 | 0.3122 | 0.4299 | 0.8037 | 0.2680 | 0.6041 | 0.4503 | 0.4187 |
| $q^*$ only | R@10 | 0.5711 | 0.5654 | 0.9102 | 0.4170 | 0.7214 | 0.5357 | 0.5342 |
| ColBERTv2 | R@3 | 0.2091 | 0.3372 | 0.7149 | 0.2580 | 0.4806 | 0.3344 | 0.3494 |
| $t^*$ only | R@10 | 0.3947 | 0.4588 | 0.8265 | 0.4060 | 0.6005 | 0.4248 | 0.4716 |
| ColBERTv2* | R@3 | **0.3592** | **0.4666** | **0.9067** | **0.3000** | **0.6862** | **0.4822** | **0.4196** |
| | R@10 | **0.6344** | **0.6018** | **0.9663** | **0.4850** | **0.7917** | **0.5694** | **0.5611** |

Table 8: Ablation study on doc-level embedding with ColBERTv2. In all cases the ensemble of relevant queries, title and chunks gives the best results. ColBERTv2* means base model plus the doc-level embedding.

## A.3 QUALITATIVE EXAMPLES ON AUGMENTED SYNTHETIC RELEVANT QUERIES UNDER DIFFERENT MODELS

| | Llama2-7B | Llama2-70B | Llama3-8B | Llama3-70B |
|---|---|---|---|---|
| User Query | A Direct Search Method to solve Economic Dispatch Problem with Valve-Point Effect | | | |
| Original Document | Dynamic economic dispatch (DED) is one of the main functions of power generation operation and control. It determines the optimal settings of generator units with predicted load demand over a certain period of time. The objective is to operate an electric power system most economically while the system is operating within its security limits. This paper proposes a new hybrid methodology for solving DED. The proposed method is developed in such a way that a simple evolutionary programming (EP) is applied as a based level search, which can give a good direction to the optimal global region, and a local search sequential quadratic programming (SQP) is used as a fine tuning to determine the optimal solution at the final. Ten units test system with nonsmooth fuel cost function is used to illustrate the effectiveness of the proposed method compared with those obtained from EP and SQP alone. | | | |

| | | | | |
|---|---|---|---|---|
| Synthetic Queries | What is dynamic economic dispatch (DED)? What is the main function of power generation operation and control? How does DED determine the optimal settings of generator units? What is the objective of DED? How does the proposed hybrid methodology for solving DED work? What is the difference between evolutionary programming (EP) and sequential quadratic programming (SQP)? What is the test system used in the article to illustrate the effectiveness of the proposed method? What is the nonsmooth fuel cost function used in the article? How does the proposed method compare with those obtained from EP and SQP alone? What is the advantage of using a hybrid methodology for solving DED? How does the proposed method improve the efficiency of DED? | What is dynamic economic dispatch (DED)? What is the objective of DED? How does the proposed hybrid methodology for DED work? What are the advantages of using the proposed hybrid methodology for DED? How does the ten units test system with nonsmooth fuel cost function illustrate the effectiveness of the proposed method? How does the proposed method compare with using only EP or SQP for DED? | What is dynamic economic dispatch (DED) in power generation operation and control? How does DED determine optimal generator settings? What is the objective of DED? What is the proposed hybrid methodology for solving DED? How does the proposed method combine evolutionary programming (EP) and sequential quadratic programming (SQP)? What is the effectiveness of the proposed method compared to EP and SQP alone? How is the proposed method used to solve DED for a 10-unit test system with nonsmooth fuel cost function? | What is dynamic economic dispatch (DED) in power generation operation and control? How does dynamic economic dispatch (DED) work? What is the objective of dynamic economic dispatch (DED)? What are the benefits of using dynamic economic dispatch (DED) in electric power systems? How to solve dynamic economic dispatch (DED) problems? What is evolutionary programming (EP) in dynamic economic dispatch (DED)? What is sequential quadratic programming (SQP) in dynamic economic dispatch (DED)? How to optimize electric power systems using dynamic economic dispatch (DED)? What are the applications of dynamic economic dispatch (DED) in power systems? What are the challenges of dynamic economic dispatch (DED) in power systems? How to determine the optimal settings of generator units in power systems? |
| User Query | Bearish-Bullish Sentiment Analysis on Financial Microblogs | | | |
| Original Document | Classifying short texts to one category or clustering semantically related texts is challenging, and the importance of both is growing due to the rise of microblogging platforms, digital news feeds, and the like. We can accomplish this classifying and clustering with the help of a deep neural network which produces compact binary representations of a short text, and can assign the same category to texts that have similar binary representations. But problems arise when there is little contextual information on the short texts, which makes it difficult for the deep neural network to produce similar binary codes for semantically related texts. We propose to address this issue using semantic enrichment. This is accomplished by taking the nouns, and verbs used in the short texts and generating the concepts and co-occurring words with the help of those terms. The nouns are used to generate concepts within the given short text, whereas the verbs are used to prune the ambiguous context (if any) present in the text. The enriched text then goes through a deep neural network to produce a prediction label for that short text representing it's category. | | | |
| Synthetic Queries | How can deep neural networks be used for classifying short texts? What are the challenges in classifying and clustering short texts? How does semantic enrichment help in addressing the challenges in classifying short texts? What is the difference between nouns and verbs in generating concepts and co-occurring words? How does the enriched text go through a deep neural network to produce a prediction label? What is the purpose of pruning ambiguous context in the text using verbs? How does the proposed method improve the accuracy of classifying short texts? What are microblogging platforms and digital news feeds? How can the proposed method be applied to real-world scenarios? What are the advantages of using semantic enrichment in text classification? How does the method handle texts with little contextual information? | How can we classify and cluster short texts? What is the challenge in classifying and clustering short texts? How do deep neural networks help in classifying and clustering short texts? What is semantic enrichment and how does it help in classifying and clustering short texts? How do we generate concepts and co-occurring words in short texts? How do we use nouns and verbs to generate concepts and prune ambiguous context in short texts? | How to classify short texts into categories? What is the importance of classifying short texts? How to cluster semantically related texts? What is the role of deep neural networks in text classification? How to address the issue of little contextual information in short texts? What is semantic enrichment in text classification? How to generate concepts and co-occurring words from nouns and verbs? How to prune ambiguous context in short texts using verbs? What is the process of predicting a category label for a short text? query: What are the applications of text classification and clustering in microblogging platforms and digital news feeds? | How to classify short texts into categories? What are the challenges of clustering semantically related texts? How can deep neural networks be used for text classification and clustering? What is semantic enrichment and how can it be used for text classification? How to handle lack of contextual information in short texts for classification? What role do nouns and verbs play in semantic enrichment for text classification? How to generate compact binary representations of short texts for classification? What are the applications of text classification and clustering in microblogging platforms and digital news feeds?" |
| User Query | When traveling to a country with a different currency, how should you take your money? | | | |
| Original Document | I think the current usual solution is to get a debit card (or failing that a credit card) with low/no foreign transaction and cash withdrawal fees. (In the UK, the Halifax Clarity Card is the best for this at the moment) Then, when you get to the country, take out cash periodically. Not too much in case of issues, but don't assume you can do it too often as you may not always be able to find a cash machine depending on where you are. It's worth getting a little bit of currency before you go (if you don't have it left over from another trip), so you can buy a coffee while you wait for everyone who got off the plane quicker than you to finish using the cash machine at the port/airport/station! Edit: Since you've said you're in the US, I can't suggest the best card for you to get to avoid/minimise foreign transaction fees, but asking on the personal finance SE site is likely the best bet for finding out what that card is. | | | |

| | | | | |
|---|---|---|---|---|
| Synthetic Queries | What are the best debit cards or credit cards for international travel with low or no foreign transaction fees? How often can I withdraw cash while traveling abroad without incurring excessive fees? Are there any prepaid travel cards that offer low or no fees for cash withdrawals? How can I minimize the fees associated with using my debit or credit card while traveling abroad? What are some tips for finding cash machines while traveling in a foreign country? Can I use my debit or credit card to get cash advances while traveling abroad? How do I avoid foreign transaction fees when using my debit or credit card while traveling? | What is the best way to manage money while traveling abroad? How can I avoid foreign transaction fees on my debit or credit card? What is the best card to use for international travel? How can I get cash while traveling abroad? Is it better to use cash or card while traveling? How can I minimize fees when using my card abroad? | How to avoid foreign transaction fees? What is the best debit card for international travel? How to get cash while traveling abroad? How to minimize cash withdrawal fees? What is the best credit card for international travel? What is the best way to exchange currency while traveling? How to find a cash machine while traveling? What are some tips for traveling abroad with a debit/credit card? | Best debit card for international travel Low foreign transaction fee credit cards Cash withdrawal fees when traveling abroad How to minimize foreign transaction fees? Best way to get cash when traveling internationally? Debit card with no foreign transaction fees? Credit card for international travel? How to avoid foreign transaction fees? Best card for international travel in the US? Minimizing cash withdrawal fees when traveling? |
| User Query | How to avoid toddlers on a long-distance plane flight? | | | |
| Original Document | For $160,000, you can hire a private jet for London - LAX return which is a similar trip length to that of London to San Francisco. This aircraft seats 13 making the trip approximately $12,300 each if you could find 12 other people to share it with. That's a similar price to paying for fully flexible first class. For that you typically get a private terminal, minimal security, ability to take almost as much luggage as you wish, and an aircraft that will wait for you and take off whenever you want. You also guarantee to be sharing the aircraft with people of your choice. The OP specifically asks "how to avoid toddlers". This question and similar questions, such as how do I avoid drunks, people who snore or sitting next to someone with bad body odor, or why should babies be allowed in first class, typically end up with the canonical answer that you should fly in a private jet. When you are taking a form of public transport, inevitably you are going to come across a disagreeable situation or disagreeable passengers. On almost any airline (apart from, I think, one exception), babies and young children are allowed in all cabins including First and Business class. Children under a certain age are not allowed in the exit row, but they are allowed in front of, behind and in the bulkhead adjacent to the exit row. There are many parents who can afford to travel in Business and First and you see young children in these every day. Some parents prefer an overnight flight to a day flight in order to get the children to sleep more quickly. In summary, you can almost never be guaranteed to more than one seat away from a toddler. If the OP had said he'd had an uncomfortable flight and asked what's the best way with dealing with noise on an aircraft, I would have provided an answer responding to that question, rather than the question that the OP did ask. | | | |
| Synthetic Queries | How much does it cost to hire a private jet for a similar trip length to London to San Francisco? What are the benefits of flying private jet compared to commercial airlines? How many people can a private jet typically seat? Are there any restrictions on who can fly on a private jet? How do I find a private jet to hire for my trip? What are some tips for flying in a private jet? How do I avoid uncomfortable situations on a commercial flight? Are there any airlines that offer private terminals and minimal security? How do I ensure a comfortable flight experience with my child? Are there any age restrictions for flying in a private jet? How do I book a private jet for my next flight? | How to avoid toddlers on a flight? How to avoid drunks on a flight? How to avoid people who snore on a flight? How to avoid sitting next to someone with bad body odor on a flight? Why are babies allowed in first class? How to guarantee a comfortable flight experience? | How to avoid toddlers on a flight? How to avoid drunks on a flight? How to avoid people who snore on a flight? Why should babies be allowed in first class? How to deal with noise on an aircraft? Can you avoid toddlers in first class? How to make a private jet trip affordable? What are the benefits of flying in a private jet? Can you choose your seatmates on a private jet? How much does it cost to hire a private jet? | How to avoid toddlers on a flight? How to avoid drunks on a plane? How to avoid people who snore on a flight? How to avoid sitting next to someone with bad body odor on a plane? Why are babies allowed in first class? How to guarantee a toddler-free flight? How to deal with noise on an aircraft? What are the benefits of flying private jet? How much does it cost to hire a private jet? Is flying private jet worth the cost? How to avoid disagreeable passengers on a flight? Can you avoid children in first class? |

Table 11: Qualitative examples on LLM generated synthetic queries under different models

| Method | Model Size | Model Architecture | Requires Training | Training FLOPS on Generated Tokens | Indexing Time FLOPS on Document Tokens | Inference FLOPS on User Query |
|---|---|---|---|---|---|---|
| Roberta + LLM Augmented Retrieval | 125M | encoder-only | No | 0 | (2*125M*1.6)x | (2*125M)x |
| RepLlama | 7B | decoder-only | Yes | (6*7B)x | (2*7B)x | (2*7B)x |
| MistralE5 | 7B | decoder-only | Yes | (6*7B)x | (2*7B)x | (2*7B)x |

Table 12: Comparison to Other Methods Using Synthetic Query

