# OpenReview forum: "LLM-Augmented Retrieval: Enhancing Retrieval Models Through Language Models and Doc-Level Embedding"
_ICLR.cc/2025/Conference — Submitted to ICLR 2025_

### Official Review · Reviewer_ZRAX · 2024-10-29

**Soundness:** 2
**Presentation:** 2
**Contribution:** 2
**Rating:** 5
**Confidence:** 4

**Summary:**

This paper proposes an LLM-augmented retrieval framework that uses a large language model (LLM), specifically LLaMA-70B, to generate pseudo-queries for document retrieval. For each document, the framework includes synthetic queries generated by the LLM, existing or LLM-generated titles, and document chunks. These text fields are embedded and combined with the query embedding using inner product, followed by weighted sum and max pooling. Evaluations are conducted with Contriever, Dragon, and ColBERT on the LoTTE dataset and a subset of the BEIR benchmark, with considerable improvements.

**Strengths:**

1. The motivation is solid, with a straightforward idea: using pseudo-queries to augment document representation, an important direction given the short context windows of current embedding models that limits their effectiveness.

2. The observed improvement appears substantial on the presented datasets.

3. The authors provide some valuable insights that could be helpful, such as the impact of titles and approaches to handle cases where titles are missing, which aligns well with real-world scenarios.

**Weaknesses:**

1. The cost of this solution appears too high, given that it relies on LLaMA-70B, and all documents require query generation. Additionally, I have concerns regarding potential data leakage in the LLM, as it may have previously encountered these query-document pairs, potentially generating similar queries based on the document. This issue could be addressed if the authors provided examples and analysis.

2. The inference process requires weighting strategies to combine the relevance between the query and different fields of the document. I have concerns about the scalability of this solution, as the presented dataset contains a relatively small number of documents (see Q1).

3. Some presentation issues: (i) The inconsistent capitalization of "Bi-encoder" is concerning, with both "Bi-encoder" and "bi-encoder" used interchangeably throughout the paper, which appears unprofessional. Additionally, on line 19, to my knowledge, ColBERTv2 falls under the category of bi-encoders but employs a more complex relevance scoring process. (ii) Certain findings in sections 3.1.2, 3.1.3, and others could be more effectively presented in an analysis section following the experiments section, providing a clearer flow and focused discussion on results.

**Questions:**

1. I am curious whether this method could be effective on larger document benchmarks, such as MSMARCO and FEVER within the BEIR benchmark. Could the authors provide results on these datasets?

2. Could the authors provide an analysis of the costs associated with using an LLM to enhance these tasks, such as the number of tokens or text checks required for generation?

3. It would be interesting to see the performance when using the original query to search against the pseudo queries, to quantify any potential leakage from the LLM (i.e., the ability to recover the query based on the document).

---

> ### Author Response · Authors · 2024-11-22
> **Rebuttal to Reviewer ZRAX**
>
> ## Rebuttal
> **[W1]** The cost of this solution appears too high, given that it relies on LLaMA-70B, and all documents require query generation. Additionally, I have concerns regarding potential data leakage in the LLM, as it may have previously encountered these query-document pairs, potentially generating similar queries based on the document. This issue could be addressed if the authors provided examples and analysis.
>
> **[A1]** Thanks for the feedback. In Section 4.5, we have included an Augmentation Analysis, highlighted in **blue** color, to discuss both cost-effectiveness and the query match ratio, Match($q^*$). This ratio is defined as the ratio of the number of intersections between search queries in the eval set and synthetic relevant queries relative to the total number of search queries in the eval set. The Match($q^*$) values are predominantly zero across all evaluation sets, except for FIQA (1.0%) and Quora (6.2%). This indicates that there is no or little data leakage, and the significant improvement in the retrieval result does not come from the data leakage. Besides, in Appendix, Table 11 provides several examples of synthetic queries generated by different LLMs.
>
> In addition, it is not necessary to use the Llama-70b model. As demonstrated in Section 4.5 and Table 11, Llama-7B and Llama-8B can perform equivalently at a lower cost. In addition, the generated tokens constitute approximately 57% of the original document size. However, these synthetic queries are pre-generated for augmentation and then pre-computed for retriever indexing only once throughout the entire process, the overall cost remains limited, particularly when the retriever model's parameter size is small (125 million).
>
> | | |Original|Documents||Generated|Synthetic|Relevant|Queries|||
> |------------|-------|--------|-----------|----------|-----------|--------------|---------|------------|-------------|----------|
> |Dataset|Subset|$N_D$(in K)|$N_{T_D}$(in K)|$N_{T_D}/N_D$|$N_{q^*}$(in K)|$N_{T_{q^*}}$(in K)|$N_{q^*}/N_D$|$N_{T_{q^*}}/N_D$|$N_{T_{q^*}}/N_{q^*}$|Match($q^*$)\%|
> |BEIR|ArguAna|9|1,782|205|46|684|5|79|15|0|
> ||FIQA|58|9,470|164|305|4,360|5|76|14|1.0|
> ||Quora|523|8,404|16|3,123|40,947|6|78|13|6.2|
> ||SciDocs|25|5,365|212|160|2,580|6|102|16|0|
> ||SciFact|5|1,548|299|32|618|6|119|19|0|
> ||CQADEnglish|40|4,251|106|179|2,987|4|74|17|0|
> ||CQADPhysics|38|6,992|182|184|3,232|5|84|18|0|
> |LoTTE|Lifestyle|119|21,639|181|664|9,866|6|83|15|0|
> ||Recreation|167|26,988|162|902|13,215|5|79|15|0|
> ||Science|1,694|400,544|236|8,461|159,901|5|94|19|0|
> ||Technology|662|117,940|178|7,031|105,610|11|159|15|0|
> ||Writing|200|29,031|145|1,027|15,364|5|77|15|0|
>
> &emsp;
>
> **[W2 & Q1]** The inference process requires weighting strategies to combine the relevance between the query and different fields of the document. I have concerns about the scalability of this solution, as the presented dataset contains a relatively small number of documents (see Q1).
> I am curious whether this method could be effective on larger document benchmarks, such as MSMARCO and FEVER within the BEIR benchmark. Could the authors provide results on these datasets?
>
> **[A2]** Thanks for the feedback. Our weights are hyperparameters determined using a dev set (BEIR-ArguAna) without extensive tuning, and they remain fixed across other evaluation sets.  We have observed that these weight selections generalize effectively across datasets (see Section 4.3, highlighted in **orange** color). Additionally, as shown in Table 4 of Section 4.5, LoTTE-Science comprises 1.7 million documents, and LoTTE-Technology contains 0.7 million documents, both of which are substantial in size. The recall performance improvements are also significant across various base models on these two evaluation sets. Consequently, we anticipate that our method will yield significant recall performance enhancements on MSMARCO and FEVER across different base models.

---

> ### Author Response · Authors · 2024-11-22
> **Rebuttal to Reviewer ZRAX - Part 2**
>
> **[W3]** Some presentation issues: (i) The inconsistent capitalization of "Bi-encoder" is concerning, with both "Bi-encoder" and "bi-encoder" used interchangeably throughout the paper, which appears unprofessional. Additionally, on line 19, to my knowledge, ColBERTv2 falls under the category of bi-encoders but employs a more complex relevance scoring process. (ii) Certain findings in sections 3.1.2, 3.1.3, and others could be more effectively presented in an analysis section following the experiments section, providing a clearer flow and focused discussion on results.
>
> **[A3]** Thanks for pointing this out. We have corrected "Bi-encoder" in our revised version and also put more analysis in Section 4.5 per your suggestion (in **blue** color).
> For ColBERT, they claim themselves as late-interaction models in their paper [1]. The primary distinction between ColBERT and a Bi-encoder lies in their handling of document indexing and query processing. A Bi-encoder can precompute its document index, requiring only the encoding of the query during inference, followed by the application of an approximate nearest neighbor (ANN) algorithm for search. In contrast, ColBERT cannot easily precompute in this manner due to its complex relevance scoring process.
>
> [1] Khattab, Omar, and Matei Zaharia. "Colbert: Efficient and effective passage search via contextualized late interaction over bert." In Proceedings of the 43rd International ACM SIGIR conference on research and development in Information Retrieval, pp. 39-48. 2020.
>
> &emsp;
>
> **[Q2]** Could the authors provide an analysis of the costs associated with using an LLM to enhance these tasks, such as the number of tokens or text checks required for generation?
>
> **[A5]** Thanks for your suggestion. We have added a new Table 4 in Section 4.5, highlighted in **blue** color, which summarizes all the statistics related to the materials generated by the large language model (LLM), including the number of queries and tokens. Additionally, our method does not require text checks during generation; instead, we simply parse the LLM outputs to construct the document-level embedding. As mentioned above, the generated tokens constitute approximately 57% of the original document size. However, as these synthetic queries are pre-generated for augmentation and then pre-computed for retriever indexing only once throughout the entire process, the overall cost remains limited, particularly when the retriever model's parameter size is small (125 million).
>
> &emsp;
>
> **[Q3]** It would be interesting to see the performance when using the original query to search against the pseudo queries, to quantify any potential leakage from the LLM (i.e., the ability to recover the query based on the document).
>
> **[A6]** As mentioned above, Table 4 in Section 4.5 includes the Match($q^*$) values (illustrating the frequency using the original query to search against the pseudo queries, which are predominantly zero across all evaluation sets, except for FIQA (1.0%) and Quora (6.2%). This indicates that there is no or very little data leakage in the LLM,  and the significant improvement in the retrieval result does not come from the data leakage

---

> > ### Comment · Reviewer_ZRAX · 2024-11-25
> >
> > Thank you for the detail response.
> >
> > For W1:
> >
> > > On average 6 synthetic relevant queries are generated per document and the token count in the generated syntheticqueries is comparable to the token count in the original documents. (l 438)
> >
> > I still believe the cost here is relatively high. Existing retrievers typically train on tens of thousands of text pairs, while this method requires generating millions of text chunks and this is not even the full extent of the cost.
> >
> > ---
> >
> > For W2:
> >
> > My concern remains. BEIR is a widely used benchmark in IR research, and I do not see a compelling reason to avoid performing tasks on other larger datasets within BEIR. In practice, million-scale corpora are quite common, and evaluating the method's performance in such scenarios would provide more valuable insights.

---

> ### Author Response · Authors · 2024-11-26
>
> Thanks for your follow-up questions.
>
> **[FQ1]**
> > I still believe the cost here is relatively high. Existing retrievers typically train on tens of thousands of text pairs, while this method requires generating millions of text chunks and this is not even the full extent of the cost.
>
> **[FA1]** We acknowledge that the generation of synthetic queries introduces an additional cost to our approach. However, several factors justify that these costs are limited and worthwhile:
>  1) The synthetic queries are generated only once and are pre-computed into the retriever index, which is stored offline. The size of the index remains consistent with other standard dense retrieval methods. Consequently, our approach does not increase latency during inference time.
>  2) Furthermore, unlike other retrievers that requires extensive training, our model does not require additional training. This enables existing retriever models, including smaller ones, to directly leverage the foundational knowledge of LLMs. As a result, training costs are reduced, and the adaptability for users is enhanced.
>  3) As demonstrated in Table 5, our analysis indicates that the generation model can be substituted with Llama 7B or Llama 8B with negligible performance degradation. This substitution further reduces the cost of generation.
>
> Specifically, smaller retriever models, such as RoBERTa, typically exhibit suboptimal performance when trained on only tens of thousands of text pairs. Most retrievers are trained on the MSMARCO dataset, which contains over 8 million passages. For instance, Contriever [1] is trained on Wikipedia and CCNet data, while DRAGON [2] undergoes iterative training on MSMARCO, followed by extensive augmentation for training and distillation. Our methodology can surpass these models without additional training by incorporating foundational knowledge from large language models (LLMs). For LLM-based architectures like RepLlama [3] and MistralE5 [4], although they possess foundational knowledge, the cost of subsequent fine-tuning is non-trivial due to their large parameter sizes. Therefore, we believe our proposed methodology strikes an optimal balance between performance and cost.
>
> In addition, inspired by another reviewer, we summarize a table to estimate the cost of our approach (using Llama 7B for generation) comparing to other LLM-retrievers like RepLlama [3] and MistralE5 [4]. Our cost is still less than the two LLM-retrievers with faster inference time (the Inference FLOPS on User Query is much smaller).
>
> |Method|Model Size|Model Architecture|Requires Training|Training FLOPS on Generated Tokens|Synthetic Query Generation + Indexing Time FLOPS on Document Tokens|Inference FLOPS on User Query|
> |-----------------|------------|-------|-------|-----------|---------|------|
> |RobertaRetriever+LLMAugmentedRetrieval|125M|encoder-only|No|0|O(2*7B)x|(2*125M)x|
> |RepLlama[3] |7B|decoder-only|Yes|(6*7B)x|(2*7B)x|(2*7B)x|
> |MistralE5[4] |7B|decoder-only|Yes|(6*7B)x|(2*7B)x|(2*7B)x|
>
> *For training FLOPS inference FLOPS and indexing time FLOPS in the table, they are estimated based on OpenAI scaling law paper [5]
>
> &emsp; &emsp;
>
> **[FQ2]**
> > My concern remains. BEIR is a widely used benchmark in IR research, and I do not see a compelling reason to avoid performing tasks on other larger datasets within BEIR. In practice, million-scale corpora are quite common, and evaluating the method's performance in such scenarios would provide more valuable insights.
>
> **[FA2]** Sure, we are evaluating the performance on MSMARCO and FEVER and will attach the performance **here** later. The results will be included in the final paper.
>
> &emsp;
>
> [1] Izacard, Gautier, Mathilde Caron, Lucas Hosseini, Sebastian Riedel, Piotr Bojanowski, Armand Joulin, and Edouard Grave. "Unsupervised dense information retrieval with contrastive learning." arXiv preprint arXiv:2112.09118 (2021).
>
> [2] Lin, Sheng-Chieh, Akari Asai, Minghan Li, Barlas Oguz, Jimmy Lin, Yashar Mehdad, Wen-tau Yih, and Xilun Chen. "How to train your dragon: Diverse augmentation towards generalizable dense retrieval." arXiv preprint arXiv:2302.07452 (2023).
>
> [3] Xueguang Ma, Liang Wang, Nan Yang, Furu Wei, and Jimmy Lin. 2023. Fine-tuning llama for multi-stage text retrieval.
>
> [4] Liang Wang, Nan Yang, Xiaolong Huang, Linjun Yang, Rangan Majumder, and Furu Wei. 2023. Improving text embeddings with large language models.
>
> [5] Kaplan, Jared, Sam McCandlish, Tom Henighan, Tom B. Brown, Benjamin Chess, Rewon Child, Scott Gray, Alec Radford, Jeffrey Wu, and Dario Amodei. "Scaling laws for neural language models." arXiv preprint arXiv:2001.08361 (2020).

---

> ### Comment · Reviewer_ZRAX · 2024-11-27
>
> Thank you for the detailed response and the effort you’ve put into addressing these points.
>
> > 1. The synthetic queries are generated only once and are pre-computed into the retriever index, which is stored offline. The size of the index remains consistent with other standard dense retrieval methods. Consequently, our approach does not increase latency during inference time.
>
> While it does not increase latency during the search phase, the overall process introduces additional overhead rather than reducing it. More experimental results are needed to justify the associated costs.
>
> > 2. Furthermore, unlike other retrievers that requires extensive training, our model does not require additional training. This enables existing retriever models, including smaller ones, to directly leverage the foundational knowledge of LLMs. As a result, training costs are reduced, and the adaptability for users is enhanced.
>
> I respectfully disagree with this point. Your method relies on a well-trained retriever as its backbone and benefits from the training it has undergone. To support the claim that no training is required, you could experiment with foundational non-retrieval models like bert-base-uncased, which I don’t think would work.
>
> A less stringent setting might involve applying your method to unsupervised retrievers like E5-base-unsupervised or Contriever (unsupervised version), or weaker retrievers like DPR, and comparing their performance after augmentation to SOTA retrievers.
>
> > In addition, inspired by another reviewer, we summarize a table to estimate the cost of our approach (using Llama 7B for generation) comparing to other LLM-retrievers like RepLlama [3] and MistralE5 [4].
>
> I agree that this addition is helpful for assessing the cost of your approach.
>
> - Could you elaborate further on the specifics? For instance, what is the precise difference between O(2x7B)x and (2x7B)x? The distinction feels counterintuitive. While they use LLMs for embedding, your approach utilizes LLMs for both generation and embedding.
>
> - Additionally, I believe the encoder-only retriever serves as a more natural baseline for comparison rather than the LLM-based retriever.
>
> > Sure, we are evaluating the performance on MSMARCO and FEVER and will attach the performance here later. The results will be included in the final paper.
>
> I would recommend evaluating on the entire BEIR benchmark for a more comprehensive assessment, as is standard practice in prior IR research.

---

> ### Author Response · Authors · 2024-11-27
>
> Thanks for your questions and appreciate the discussion here. First of all, we'd like to briefly summarize our proposed method here to avoid any miscommunication or confusion.
> 1) Given an pre-existing retriever model and a foundational LLM
> 2) Use the LLM to generate query field and title field (optional). LLM is **ONLY** applied at this step.
> 3) We use the existing retriever model to compute the chunk embedding, query embedding, title embedding and therefore calculate the doc-level embedding described in eq. (1), and store them to construct the retriever index. The total number of retriever index does not increase.
> 4) Inference is the same as the standard dense retrieval process. Specifically, only the given retrieval model is employed to compute the user query embedding. From this point onward, the procedure follows the conventional retrieval setup.
> 5) Note that the given retrieval model is not fine-tuned or further trained at all.
>
> &emsp;
>
> > While it does not increase latency during the search phase, the overall process introduces additional overhead rather than reducing it. More experimental results are needed to justify the associated costs.
>
> Yes we acknowledge the additional overhead here. Our paper seeks to propose a model-agnostic framework that enhances the recall performance of existing retrievers without requiring further training (fine-tuning), while recognizing that the trade-off involves the number of tokens generated. We would appreciate it if you could specify the types of experiments you are interested in here.
>
> &emsp;
>
> > I respectfully disagree with this point. Your method relies on a well-trained retriever as its backbone and benefits from the training it has undergone. To support the claim that no training is required, you could experiment with foundational non-retrieval models like bert-base-uncased, which I don’t think would work.
> A less stringent setting might involve applying your method to unsupervised retrievers like E5-base-unsupervised or Contriever (unsupervised version), or weaker retrievers like DPR, and comparing their performance after augmentation to SOTA retrievers.
>
> Here we mean no additional training or fine-tuning will be needed. Practically, users often continue to fine-tune their retrieval models using augmented datasets, which may include data generated by LLMs. The trade-off here involves balancing the cost-effectiveness of token generation against improvements in recall performance. Alternatively, to enhance recall performance, users may need to either continue training their retrieval models or employ a larger base model, which might also necessitate ongoing training.
>
> &emsp;
>
> > I agree that this addition is helpful for assessing the cost of your approach.
> Could you elaborate further on the specifics? For instance, what is the precise difference between O(2x7B)x and (2x7B)x? The distinction feels counterintuitive. While they use LLMs for embedding, your approach utilizes LLMs for both generation and embedding.
> Additionally, I believe the encoder-only retriever serves as a more natural baseline for comparison rather than the LLM-based retriever.
>
> O(2x7B)x $\approx$ (2x7B)x (for token generation) + (2x125Mx1.6)x (for encoding tokens)
>
> 1.6 includes the ~60% more tokens we generated. To be clear our approach **ONLY** utilize LLMs for generation (**NOT** for embedding).
>
> Furthermore, the objective of this paper is to utilize the foundational knowledge inherent in lLLMs to enhance existing recall performance. Employing LLM-based retrievers represents an alternative approach to improving retriever performance by capitalizing on the foundational knowledge of LLMs, and we include this as a point of comparison.
>
> Moreover, I recommend that we concentrate on the recall performance of our proposed method, as we do not assert that our approach is without cost. Our current discussion appears to be diverging from the primary aim of the paper. In practical terms, this method is particularly well-suited for enterprise search applications where the search domain is restricted and high recall performance is essential.
>
> &emsp;
>
> > I would recommend evaluating on the entire BEIR benchmark for a more comprehensive assessment, as is standard practice in prior IR research.
>
> Sure, we'll include the entire BEIR benchmark then. But given the limited time and compute resources we have now, we probably will attach here first and update in the final revision of the paper.
>
> &emsp;
>
> Furthermore, if our actions and responses have satisfactorily addressed your questions and concerns, we would greatly appreciate it if you could consider increasing your rating of our paper. Your support would be highly valued!

---

> > ### Comment · Reviewer_ZRAX · 2024-11-27
> >
> > Thanks for clarifcation.
> >
> > > The distinction feels counterintuitive. While they use LLMs for embedding, your approach utilizes LLMs for both generation and embedding.
> >
> > Apologies for the mistake in my initial comment. I understand that your approach employs LLMs for generation only, with the subsequent retrieval process following a conventional retriever pipeline. My points are
> > - It is unfair to emphasize your effectiveness over encoder-only retrievers like Contriever while comparing your costs to LLM-based retrievers like RepLlama.
> > - The generation cost is several orders of magnitude higher than the embedding cost, making it somewhat misleading to use O() notation to omit such a significant disparity.
> > - The cost is essentially on par with using LLMs to construct synthetic datasets tailored to the corpus and task. A comparable baseline might involve employing a retriever trained on such synthetic datasets.
> >
> >
> > To clarify, I am not advocating for a strictly low-cost approach or imposing overly harsh standards on cost. In fact, I would appreciate and commend the use of LLMs with increased cost if they significantly enhance effectiveness. **However, I feel that your comparison is unfairly presented and appears biased toward supporting your claim that your method is cost-effective.**

---

> > > ### Author Response · Authors · 2024-11-27
> > >
> > > Thanks for your comment. We appreciate your perspective on evaluating cost-effectiveness. However, it is important to note that assessing cost-effectiveness is inherently subjective and, more importantly, is not the main claim of our paper. We would be grateful if we could shift our focus to other aspects of the study and would be pleased to address any additional questions you may have.

---

> > > > ### Comment · Reviewer_ZRAX · 2024-11-27
> > > >
> > > > Thank you for your response. Over the past two years, there’s been a noticeable rise in works leveraging LLMs for tasks like data labeling or augmentation. I think it’s really important to clearly explain why LLMs are being used and to be upfront about the costs and benefits involved—it’s just getting a bit tiring seeing this overlooked.
> > > >
> > > > To clarify: I do not think it is necessary to aim for the lowest cost with the highest gain for acceptance, but it is vital to ensure that comparisons are **fair** and **well-justified**. Many of the concerns and discussions raised above could be more effectively addressed after you present the BEIR and MSMARCO benchmarks. I look forward to seeing the results.

---

> > > > > ### Author Response · Authors · 2024-12-04
> > > > >
> > > > > Please refer to the table in the main section for results on BEIR and nDCG@10.

---

### Official Review · Reviewer_fyxe · 2024-11-03

**Soundness:** 3
**Presentation:** 3
**Contribution:** 2
**Rating:** 5
**Confidence:** 4

**Summary:**

This paper proposes a novel retrieval framework called to improve the performance of current retrieval models. The framework achieves this by enriching document embeddings with information from large language models (LLMs), such as synthetic queries and titles. The proposed document-level embedding approach integrates this augmented information with existing document chunks, and it's shown to be effective for both bi-encoder and late-interaction retrieval models. The authors demonstrate the framework's effectiveness by achieving state-of-the-art results on benchmark datasets LoTTE and BEIR.

**Strengths:**

The paper is clear and the results are good.

**Weaknesses:**

It's a rather straightforward model. The novelty is simply calling some LLM to generate titles and queries for each doc, which the retriever will later use to retrieve the document. I think ICLR requires more model novelty.

**Questions:**

All clear to me.

---

> ### Author Response · Authors · 2024-11-22
> **Rebuttal to Reviewer fyxe**
>
> ## Rebuttal
>
> **[W1]** It's a rather straightforward model. The novelty is simply calling some LLM to generate titles and queries for each doc, which the retriever will later use to retrieve the document. I think ICLR requires more model novelty.
>
> **[A1]** Thanks for the feedback. To the best of our knowledge, most embedding-based retrieval methods primarily focus on computing embeddings from the original content, such as chunks of the original document, with synthetic queries predominantly used for training purposes. In contrast, our approach introduces several novel aspects:
> 1. We present a **training-free**, straightforward yet effective method that demonstrates improvements over existing state-of-the-art (SoTA) retrievers.
> 2. Our approach maintains inference speed without any sacrifice, and the total cost is minimized as the retriever index is pre-computed only once prior to the inference step. Additionally, the index size remains consistent with that of single-vector dense retrieval while including richer information from multiple fields.
> 3. We provide valuable insights, such as the impact of titles on information retrieval and the corresponding methods to employ when titles are absent.
> 4. Our solution is **model-agnostic** (also multilingual-agnostic) and can be applied to both Bi-encoder and token-level late-interaction models.

---

> > ### Author Response · Authors · 2024-11-27
> >
> > Hi Reviewer fyxe,
> >
> > We would like to confirm whether our response adequately addresses your concerns regarding novelty. Thank you very much.

---

### Official Review · Reviewer_z7Jy · 2024-11-04

**Soundness:** 2
**Presentation:** 3
**Contribution:** 3
**Rating:** 8
**Confidence:** 3

**Summary:**

The paper presents an approach to document expansion using LLMs to enhance the zero-shot retrieval effectiveness of existing state-of-the-art bi-encoder retrievers (Contriever, DRAGON and ColBERTv2).

**Strengths:**

1. The proposed method is straightforward and easy to implement and shows significant improvements for existing state-of-the-art retrievers.
2. The proposed approach is useful, which combines the document embedding from different sources and forms a single document embedding. Thus, the index size is still the same for single-vector dense retrieval.

**Weaknesses:**

1. Although the proposed approach is simple and effective, I think the major concern for me is the limited discussion in ablations and comparisons, which are important for the readers who want to adopt this approach. I'm willing to raise score if the authors can address the concern below: (1) As a reader, one may want to know the quality and cost tradeoff of the generated titles and queries. For instance, the paper prompts Llama-70B to make it work; how about using smaller size of Llama; 7B, 13B etc.? and the additional cost of document indexing with LLM document expansion should be reported. (2) There are no ablations on the three combinations: chunk + query, chunk + title and query + title, which helps readers to choose the combinations with a balanced tradeoff between effectiveness and cost  (3) A comparison with LLM-based retrievers, such as RepLlama[1] or MistralE5[2], in terms of retrieval effectiveness, query latency and indexing time, can provide more useful information for readers to choose which approach to use. (4) Can we apply this approach to multilingual retrieval tasks?
2. Although the proposed method show improvements over existing SoTA retrievers, the proposed approach is a bit incremental. I think the approach to augment the document is very similar to the previous work[3][4]. The authors should make a comparison with the previous work in Related Work to make the contribution more clear.
3. Some experimental details are missing (see Questions).

[1] Xueguang Ma, Liang Wang, Nan Yang, Furu Wei, and Jimmy Lin. 2023. Fine-tuning llama for multi-stage text retrieval.
[2] Liang Wang, Nan Yang, Xiaolong Huang, Linjun Yang, Rangan Majumder, and Furu Wei. 2023. Improving text embeddings with large language models.
[3] Rodrigo Nogueira and Jimmy Lin. 2019. From doc2query to docTTTTTquery.
[4] Yongqi Li, Nan Yang, Liang Wang, Furu Wei, and Wenjie Li. 2023. Multiview identifiers enhanced generative retrieval.

**Questions:**

1. Clarified on implementation details: according to Figure1, for each passage coming from the same document, it seems you use the same synthetic queries and titles? However, in the experiments, you mention you use the original chunk from the datasets; then, how do you know which chunks are coming from the same document and use the document to generate synthetic queries and titles for those chunks.
2. Which dev set are used to tune the hyperparameters, $w_{query}, w_{title}, w_{chunk}$?
3. From the prompts, it seems that the models are instructed to output multiple relevant queries; however, only one generated query is used for document expansion. How do you choose the query among all the generated ones? Or if you use multiple generated queries, how do you combine them?

---

> ### Author Response · Authors · 2024-11-22
> **Rebuttal to Reviewer z7Jy**
>
> ## Rebuttal
> **[W1]** Although the proposed approach is simple and effective, I think the major concern for me is the limited discussion in ablations and comparisons, which are important for the readers who want to adopt this approach. I'm willing to raise score if the authors can address the concern below:
>
> **[A1**] Thanks for the great feedback. We have included more discussions and Augmentation Analysis in Section 4.5. Per your questions:
>
> **[W1.1]** As a reader, one may want to know the quality and cost tradeoff of the generated titles and queries. For instance, the paper prompts Llama-70B to make it work; how about using smaller size of Llama; 7B, 13B etc.? and the additional cost of document indexing with LLM document expansion should be reported.
>
> **[A1.1]** We have experimented with Llama2-7B, Llama2-70B, Llama3-8B, and Llama3-70B for synthetic query generation. However, these models do not produce significant differences in the generated synthetic queries. Several qualitative examples are provided in Table 11 in the **Appendix**, highlighted in **blue** color. Although we chose 70B in the full experiments, we can also utilize Llama2-7B or Llama3-8B for synthetic query generation to improve cost-effectiveness in practical applications. Moreover, in Table 4, Section 4.5, we present a cost analysis of our method, indicating that, on average, 57% more tokens are generated compared to the original document tokens. This may result in a 57% increase in encoding costs when building the retriever index. Nevertheless, since retriever models are typically small (125 million parameters) and the retriever index is pre-computed once and stored (with no additional storage costs as the number of retriever indexes remains unchanged in our approach), the overall cost remains limited.
>
> &emsp;
>
> **[W1.2]** There are no ablations on the three combinations: chunk + query, chunk + title and query + title, which helps readers to choose the combinations with a balanced tradeoff between effectiveness and cost
>
> **[A1.2]** Thanks for the suggestion. The effectiveness of these combinations (chunk + query, chunk + title, and query + title) can be assessed by comparing our each of our ablation study results (with only single field chunk, query or title) with the results when three fields are all present (the last row) to measure the delta effects (Tables 5\~7). Furthermore, the chunk component is essential to our framework, particularly for Bi-encoders, as our method is based on the traditional query-chunk embedding similarity system [1,2,3]. The cost associated with titles is negligible, as they typically comprise only a few tokens per document. The primary cost factor is the synthetic query component, where the number of generated tokens averages 57% of the original document tokens (see Section 4.5). However, in our ablation study we find out that synthetic queries generally play a pivotal role in enhancing recall performance so we'd better keep them in our doc-level embedding. It is important to note that since the token generation and retriever index building occur only once throughout the entire process, the overall cost impact is limited.
>
> [1] Chen, Tong, Hongwei Wang, Sihao Chen, Wenhao Yu, Kaixin Ma, Xinran Zhao, Dong Yu, and Hongming Zhang. "Dense x retrieval: What retrieval granularity should we use?." arXiv preprint arXiv:2312.06648 (2023).
>
> [2] Finardi, Paulo, Leonardo Avila, Rodrigo Castaldoni, Pedro Gengo, Celio Larcher, Marcos Piau, Pablo Costa, and Vinicius Caridá. "The Chronicles of RAG: The Retriever, the Chunk and the Generator." arXiv preprint arXiv:2401.07883 (2024).
>
> [3] Lewis, Patrick, Ethan Perez, Aleksandra Piktus, Fabio Petroni, Vladimir Karpukhin, Naman Goyal, Heinrich Küttler et al. "Retrieval-augmented generation for knowledge-intensive nlp tasks." Advances in Neural Information Processing Systems 33 (2020): 9459-9474.

---

> ### Author Response · Authors · 2024-11-22
> **Rebuttal to Reviewer z7Jy - Part 2**
>
> **[W1.3]** A comparison with LLM-based retrievers, such as RepLlama[1] or MistralE5[2], in terms of retrieval effectiveness, query latency and indexing time, can provide more useful information for readers to choose which approach to use.
>
> **[A1.3]** We summarize the comparison in a table below and also included it in Table 11 with **violet** color in **Appendix** due to the page limit. For training FLOPS. inference FLOPS, and indexing time FLOPS, they are estimated based on OpenAI scaling law paper [1] which uses $C_\text{forward} \approx 2N$ and $C_\text{forward+backward} \approx 6N$. We can see both RepLlama and MistralE5 incur significantly higher costs due to the necessity of training on synthetic tokens and their substantially larger model parameters. Additionally, the query latency for RepLlama and MistralE5 is elevated because of the increased number of FLOPs required by their larger model sizes. Despite the need to encode approximately 0.6 times more tokens (as detailed in Section 4.5, where we generate 57% more tokens) to construct the retriever index, our method still results in lower indexing time FLOPs compared to RepLlama and MistralE5, due to their larger model size.
>
> |Method|Model Size|Model Architecture|Requires Training|Training FLOPS on Generated Tokens|Indexing Time FLOPS on Document Tokens|Inference FLOPS on User Query|
> |-----------------|------------|----------|-------|------------|---------|------|
> |RobertaRetriever+LLMAugmentedRetrieval|125M|encoder-only|No|0|(2\*125M*1.6)x|(2*125M)x|
> |RepLlama|7B|decoder-only|Yes|(6*7B)x|(2*7B)x|(2*7B)x|
> |MistralE5|7B|decoder-only|Yes|(6*7B)x|(2*7B)x|(2*7B)x|
>
> *For training FLOPS inference FLOPS and indexing time FLOPS in the table, they are estimated based on OpenAI scaling law paper[1]
>
> [1] Kaplan, Jared, Sam McCandlish, Tom Henighan, Tom B. Brown, Benjamin Chess, Rewon Child, Scott Gray, Alec Radford, Jeffrey Wu, and Dario Amodei. "Scaling laws for neural language models." arXiv preprint arXiv:2001.08361 (2020).
>
> &emsp;
>
> **[W1.4]** Can we apply this approach to multilingual retrieval tasks?
>
> **[A1.4]** Yes, our approach is model-agnostic and language-agnostic. It utilizes an encoder model to compute text embeddings, and as long as this foundational encoder model is capable of handling multilingual embeddings, such as XLM-RoBERTa, our approach can be effectively applied to multilingual retrieval tasks.
>
> &emsp;
>
> **[W2]** Although the proposed method shows improvements over existing SoTA retrievers, the proposed approach is a bit incremental. I think the approach to augment the document is very similar to the previous work[3][4]. The authors should make a comparison with the previous work in Related Work to make the contribution more clear.
>
> **[A2]**  Thanks for the suggestions. We have added our contributions clearer in Related Work in **violet** color and also included these two citations. In summary, prior research primarily employs a fine-tuned model for query generation [4] or utilizes generated queries for training retriever models [3]. In contrast, our approach is training-free, requiring no fine-tuning, and leverages the foundational knowledge of LLMs for query generation, as well as the foundational knowledge of retrievers for calculating similarity scores. By eliminating the need for training, we can minimize costs and ensure that the method generalizes effectively across various scenarios.
>
> [3] Rodrigo Nogueira and Jimmy Lin. 2019. From doc2query to docTTTTTquery.
>
> [4] Yongqi Li, Nan Yang, Liang Wang, Furu Wei, and Wenjie Li. 2023. Multiview identifiers enhanced generative retrieval.
>
> &emsp;
>
> **[Q1]** Clarified on implementation details: according to Figure1, for each passage coming from the same document, it seems you use the same synthetic queries and titles? However, in the experiments, you mention you use the original chunk from the datasets; then, how do you know which chunks are coming from the same document and use the document to generate synthetic queries and titles for those chunks.
>
> **[A3]** Thanks for your question. The chunks are extracted and chopped from the document, establishing a known many-to-one (M-to-1) relationship between chunks and their corresponding documents. You are correct in noting that we use the same synthetic queries and titles for each document across all corresponding chunks, as we use the entire document to generate these queries and titles. Furthermore, in Equation (1) of Section 3.2.1, all queries and titles are aggregated into a single embedding for each document and represented as $s(q, d)$, thereby maintaining a deterministic many-to-one relationship between the chunk embeddings and the combined embeddings of queries and titles.

---

> > ### Comment · Reviewer_z7Jy · 2024-11-25
> >
> > Thanks for the the response. I think the remaining concern for me is A1.3, where I think the index costs are not compared fairly. I think this main cost for this approach is to generate queries and title expansion using Llama70B; however, in the table in A1.3, the flops for the proposed approach is only from encoding the whole text to embedding but without counting query and title expansion, which I think is the key tradeoff when a user would consider when applying this approach. I've raised the score but I think these number should be correctly reported or correct if I'm wrong.

---

> > > ### Author Response · Authors · 2024-11-25
> > > **Rebuttal to z7Jy**
> > >
> > > Thanks for your question! Yes, in Table A1.3, the costs of query (and title) generation are excluded, as we assume the same source of synthetic queries (and titles) across all three scenarios. This allows us to focus on comparing the remaining costs presented in the table.
> > >
> > > We acknowledge that the majority of the expenses are likely to arise from query (and titl)e generation. Should we include these costs, they would be equivalent to (2\*70B)x for all scenarios, assuming the use of the Llama-70B model for query generation. However, as indicated in A1.1, our findings suggest minimal differences in query augmentation when utilizing either the 70B or 7B models. Consequently, the cost of query generation could potentially be reduced to (2\*7B)x in terms of tokens generated. If we use 7B model for query generation, then the table above will become below (highlighted in bold).
> > >
> > > |Method|Model Size|Model Architecture|**Synthetic Queries Generation**|Requires Training|Training FLOPS on Generated Tokens|Indexing Time FLOPS on Document Tokens|Inference FLOPS on User Query|
> > > |-----------------|------------|----------|-------|-------|-----------|---------|------|
> > > |RobertaRetriever+LLMAugmentedRetrieval|125M|encoder-only|**(2*7B)x**|No|0|(2\*125M*1.6)x|(2*125M)x|
> > > |RepLlama|7B|decoder-only|**(2*7B)x**|Yes|(6*7B)x|(2*7B)x|(2*7B)x|
> > > |MistralE5|7B|decoder-only|**(2*7B)x**|Yes|(6*7B)x|(2*7B)x|(2*7B)x|

---

> > > > ### Comment · Reviewer_z7Jy · 2024-11-25
> > > >
> > > > Thanks. Please make the claim explicitly which I think is the performance tradeoff for the approach ``We acknowledge that the majority of the expenses are likely to arise from query (and titl)e generation. Should we include these costs, they would be equivalent to (2*70B)x for all scenarios, assuming the use of the Llama-70B model for query generation.''. However, when mentioning that it can be replaced with Llama 7B with almost no performance degrade, do you report the performance of using Llama7B as expansion model somewhere in the paper? I did not see the results.

---

> ### Author Response · Authors · 2024-11-22
> **Rebuttal to Reviewer z7Jy - Part 3**
>
> **[Q2]** Which dev set are used to tune the hyperparameters $w_{query}$, $w_{title}$, $w_{chunk}$?
>
> **[A4]** Thanks for the question. We use the dev set of BEIR-ArguAna to choose all the hyperparameters (though not heavily tuned) and fix the hyperparameters across all the evaluation sets. The hyperparameters seem to generalize really well. In addition,  $w_{query}$, $w_{title}$, $w_{chunk}$ are changed to $w_{q^*}$, $w_{t^*}$, $w_c$ for better presentation.
>
> &emsp;
>
> **[Q3]** From the prompts, it seems that the models are instructed to output multiple relevant queries; however, only one generated query is used for document expansion. How do you choose the query among all the generated ones? Or if you use multiple generated queries, how do you combine them?
>
> **[A5]** Sorry for the confusion here. Actually all synthetic queries are utilized in the calculation of the document-level embedding. These queries are firstly encoded and subsequently averaged into a single embedding vector, which is then integrated into the document embedding. We have improved Equation (1) and (2) in **teal** color in Section 3.2.1 for better presentation.

---

> ### Author Response · Authors · 2024-11-26
>
> Thanks for the follow up question. We have included Table 5 and Table 11 in Section 4.6.1 to indicate that our method can be replaced with Llama2-7b or Llama3-8b with almost no performance degrade (see attached below). In some cases our Llama2-7b and Llama3-8b even outperform Llama2-70b model.
>
>
> | Model      | Dataset | Metrics | Llama2-7b | Llama2-70b | Llama3-8b | Llama3-70b |
> |------------|---------|---------|-----------|------------|-----------|------------|
> | Contriever* | Arguana | R@3     | 0.2425    | 0.2468     | 0.2447    | 0.2596     |
> |            |         | R@10    | 0.5583    | 0.5825     | 0.5939    | 0.6110     |
> | Contriever* | Scifact | R@3     | 0.5870    | 0.5996     | 0.5996    | 0.6231     |
> |            |         | R@10    | 0.7106    | 0.7259     | 0.7196    | 0.7430     |
> | Dragon*     | Arguana | R@3     | 0.4132    | 0.3663     | 0.4232    | 0.4289     |
> |            |         | R@10    | 0.7269    | 0.6764     | 0.7496    | 0.7624     |
> | Dragon*     | Scifact | R@3     | 0.6303    | 0.6610     | 0.6348    | 0.6528     |
> |            |         | R@10    | 0.7520    | 0.7710     | 0.7538    | 0.7592     |
>
> In addition, if we combine the query generation and indexing FLOPs into one column (to make the table cleaner), it will become:
> |Method|Model Size|Model Architecture|Requires Training|Training FLOPS on Generated Tokens|Synthetic Query Generation + Indexing Time FLOPS on Document Tokens|Inference FLOPS on User Query|
> |-----------------|------------|-------|-------|-----------|---------|------|
> |RobertaRetriever+LLMAugmentedRetrieval|125M|encoder-only|No|0|O(2*7B)x|(2*125M)x|
> |RepLlama|7B|decoder-only|Yes|(6*7B)x|(2*7B)x|(2*7B)x|
> |MistralE5|7B|decoder-only|Yes|(6*7B)x|(2*7B)x|(2*7B)x|

---

> ### Comment · Reviewer_z7Jy · 2024-11-27
>
> Thanks for the response. After reading other reviewers' concerns, I also agree that NDCG@10 should be reported (I'm not sure if this approach can improve NDCG@10 consistently overall the datasets since usually we observe a tradeoff between ranking accuracy, such as NDCG@10 and MRR@10, and recall). If no consistent improvement on NDCG@10 in all the datasets, the paper should explicitly mention that the approach is mainly improving recall of the retrieval but still report NDCG@10 for reference, which I think is also important for readers who want to consider before adopting your approach. If this is the case (I'm not asking for new experiments at this stage), it would be better to conduct reranking comparison over the retrieved candidates from the models with and without the expansion in the future so that the main benefits of the approach (i.e., improving recall) can be clearly demonstrated.

---

> ### Author Response · Authors · 2024-11-27
>
> Sure, thanks for your comments. We'll include NDCG@10 here later after we've calculated NDCG@10 on BEIR sets, (and in the final revision of the paper since the paper deadline is approaching).

---

> > ### Author Response · Authors · 2024-12-04
> >
> > Please refer to the table in the main section for results on BEIR and nDCG@10.

---

### Official Review · Reviewer_nf7F · 2024-11-05

**Soundness:** 2
**Presentation:** 3
**Contribution:** 3
**Rating:** 5
**Confidence:** 4

**Summary:**

This paper introduces a novel, model-agnostic approach for document embedding, leveraging an augmented document field comprising synthetically generated queries, titles, and document chunks. With a new similarity function, the method computes a similarity score between the query and document field as a proxy for query-document relevance. The method outperforms using naive document embeddings from three existing retrievers (Contriever, DRAGON, ColBERTv2) on two benchmarks (BeIR, LoTTE).

**Strengths:**

- I like the idea of enhancing document embeddings prior to the inference step. Specifically, this can be pre-computed, offering practical advantages over query expansion methods which have been studied extensively.
- Proposed method can be applied to both bi-encoder and token-level late-interaction models, and it achieves consistently strong results across most datasets.

**Weaknesses:**

- There lacks detailed analysis of how (synthetically generated) document fields help the retrieval.  For example, how many queries were generated per document on average? The limitations section mentions that the augmented text may be as long as the original text, which suggests a substantial number of queries—clarification here would be valuable. Also, could you include some qualitative examples?
- In the ablation study, weights for each field are exclusively set to 1.0. However, the “left term” at equation (1) remains active, meaning the query-chunks still affect the results. This raises concerns about whether the ablation study truly isolates the impact of each field, especially for the query and title.
- The results are evaluated only on the subset of BeIR benchmark without any explanations.

**Questions:**

- How was the chunk size determined?
- I suggest a retouch on line 348-350 “augmentation of document embeddings with LLMs can substantially elevate the recall performance of retrieval models without necessitating additional fine-tuning”. I don’t think one model is simply fine-tuned relative to another, but it’s their complexity that differs.
- Could you cite some prior work on chunk-level embeddings (line 246-248)? Using a single document-level embedding, such as truncating, seems to be a common practice. In such cases, the proposed method will increase the document index size by |doc_len| / |chunk_size|.

---

> ### Author Response · Authors · 2024-11-22
> **Rebuttal to Reviewer nf7F**
>
> ## Rebuttal
> **[W1]** There lacks detailed analysis of how (synthetically generated) document fields help the retrieval. For example, how many queries were generated per document on average? The limitations section mentions that the augmented text may be as long as the original text, which suggests a substantial number of queries—clarification here would be valuable. Also, could you include some qualitative examples?
>
> **[A1]** Thanks for the great feedback. We have incorporated a quantitative summary table in Section 4.3 in **blue** color, which includes additional information such as $N_{q^*}/N_D$ (the average number of queries generated per document), $N_{T_{q^*}}/N_D$ (the average number of generated tokens per document), $N_{T_D}/N_D$ (the average number of tokens in the original document).
>
> Generally, the value of $N_{q^*}/N_D$ is approximately 6, indicating that, on average, six synthetic questions are generated per document. Additionally, the average $N_{T_Q}/N_D$ is 57%, suggesting that the generated tokens slightly exceed half of the original tokens. Furthermore, we have included qualitative examples in Table 11 within the **Appendix** (due to page size constraints) to compare the quality of generated queries by different large language models (LLMs) and to illustrate how these synthetic queries can enhance the informational content of the documents.
>
> | | |Original|Documents||Generated|Synthetic|Relevant|Queries|||
> |------------|-------|--------|-----------|----------|-----------|--------------|---------|------------|-------------|----------|
> |Dataset|Subset|$N_D$(in K)|$N_{T_D}$(in K)|$N_{T_D}/N_D$|$N_{q^*}$(in K)|$N_{T_{q^*}}$(in K)|$N_{q^*}/N_D$|$N_{T_{q^*}}/N_D$|$N_{T_{q^*}}/N_{q^*}$|Match($q^*$)\%|
> |BEIR|ArguAna|9|1,782|205|46|684|5|79|15|0|
> ||FIQA|58|9,470|164|305|4,360|5|76|14|1.0|
> ||Quora|523|8,404|16|3,123|40,947|6|78|13|6.2|
> ||SciDocs|25|5,365|212|160|2,580|6|102|16|0|
> ||SciFact|5|1,548|299|32|618|6|119|19|0|
> ||CQADEnglish|40|4,251|106|179|2,987|4|74|17|0|
> ||CQADPhysics|38|6,992|182|184|3,232|5|84|18|0|
> |LoTTE|Lifestyle|119|21,639|181|664|9,866|6|83|15|0|
> ||Recreation|167|26,988|162|902|13,215|5|79|15|0|
> ||Science|1,694|400,544|236|8,461|159,901|5|94|19|0|
> ||Technology|662|117,940|178|7,031|105,610|11|159|15|0|
> ||Writing|200|29,031|145|1,027|15,364|5|77|15|0|
>
> &emsp;
>
> **[W2]** In the ablation study, weights for each field are exclusively set to 1.0. However, the “left term” at equation (1) remains active, meaning the query-chunks still affect the results. This raises concerns about whether the ablation study truly isolates the impact of each field, especially for the query and title.
>
> **[A2]** Thanks for the great feedback. The purpose of the ablation study presented here is to specifically evaluate the impact of the "right term", $s(q, d)$ in equation (1), as the "left term" pertains to the conventional query-chunk embedding similarity. For clarity, we have rewritten this term in **teal** color. Furthermore, since the influence of synthetic queries and titles is confined to $s(q, d)$, we can ensure that the effects of queries and titles are isolated in this ablation study, thereby focusing on their impact on the similarity between the query and the document.
>
> &emsp;
>
> **[W3]** The results are evaluated only on the subset of the BeIR benchmark without any explanations.
>
> **[A3]** Actually our results are evaluated not only on BEIR but also on the LoTTE dataset. The selection of BEIR and LoTTE is due to their widespread adoption in both the DRAGON [1] and Colbert [2] papers. We aim to demonstrate that our LLM-augmented retrieval method can achieve advancements over these SoTA models in their corresponding evaluation sets. In addition, we randomly selected subsets from the BEIR dataset and observed significant improvements in recall performance using our method. Therefore, we believe it is unnecessary to include all evaluation sets within BEIR.
>
> [1] Lin, Sheng-Chieh, Akari Asai, Minghan Li, Barlas Oguz, Jimmy Lin, Yashar Mehdad, Wen-tau Yih, and Xilun Chen. "How to train your dragon: Diverse augmentation towards generalizable dense retrieval." arXiv preprint arXiv:2302.07452 (2023).
> [2] Khattab, Omar, and Matei Zaharia. "Colbert: Efficient and effective passage search via contextualized late interaction over bert." In Proceedings of the 43rd International ACM SIGIR conference on research and development in Information Retrieval, pp. 39-48. 2020.
>
> &emsp;
>
> **[Q1]** How was the chunk size determined?
>
> **[A4]** In our experiment, we set the chunk size to 64 tokens for Bi-encoders as a hyperparameter. This chunk size, along with other hyperparameters, was selected based on the dev set of BEIR-ArguAna, although these parameters were not extensively tuned. We then applied these settings consistently across all datasets, and they appear to generalize well across various evaluation sets.

---

> ### Author Response · Authors · 2024-11-22
> **Rebuttal to Reviewer nf7F - Part 2**
>
> **[Q2]** I suggest a retouch on line 348-350 “augmentation of document embeddings with LLMs can substantially elevate the recall performance of retrieval models without necessitating additional fine-tuning”. I don’t think one model is simply fine-tuned relative to another, but it’s their complexity that differs.
>
> **[A5]** Thanks for the great feedback. Sorry for the confusion here. Our intention was to convey that, in comparison to the base candidates (the same base model without augmented document embeddings), augmentation with LLMs can significantly enhance recall performance. Specifically, the augmentation alone can improve retrieval results without the need for retraining or fine-tuning the base model. However, including this statement here may have led to misunderstandings, as readers might interpret it as a comparison between different models. Therefore, we have removed this sentence to avoid any ambiguity.
>
> &emsp;
>
> **[Q3]** Could you cite some prior work on chunk-level embeddings (line 246-248)? Using a single document-level embedding, such as truncating, seems to be a common practice. In such cases, the proposed method will increase the document index size by |doc_len| / |chunk_size|.
>
> **[A6]** Thanks for the great feedback. Sure I’ve added citations in Section 3.1.3 in **purple** color and also listed below. Actually document chunking is a widely utilized technique in the fields of information retrieval and retrieval-augmented generation (RAG) [1,2,3]. Companies such as LangChain and LlamaIndex have developed tools to facilitate semantic chunking. The traditional method of chunk-level embedding typically increases the retriever index size by a factor of |doc_len| / |chunk_size|, and it is a standard approach in industry. Our research builds upon this conventional chunk-level embedding method, thereby not adding additional retriever indexes to the system, while significantly enhancing recall performance.
>
> [1] Chen, Tong, Hongwei Wang, Sihao Chen, Wenhao Yu, Kaixin Ma, Xinran Zhao, Dong Yu, and Hongming Zhang. "Dense x retrieval: What retrieval granularity should we use?." arXiv preprint arXiv:2312.06648 (2023).
>
> [2] Finardi, Paulo, Leonardo Avila, Rodrigo Castaldoni, Pedro Gengo, Celio Larcher, Marcos Piau, Pablo Costa, and Vinicius Caridá. "The Chronicles of RAG: The Retriever, the Chunk and the Generator." arXiv preprint arXiv:2401.07883 (2024).
>
> [3] Lewis, Patrick, Ethan Perez, Aleksandra Piktus, Fabio Petroni, Vladimir Karpukhin, Naman Goyal, Heinrich Küttler et al. "Retrieval-augmented generation for knowledge-intensive nlp tasks." Advances in Neural Information Processing Systems 33 (2020): 9459-9474.

---

> > ### Comment · Reviewer_nf7F · 2024-11-25
> > **Reply**
> >
> > Thank you for your reply, which has addressed W1, Q1, Q2. Regarding Q3, I appreciate it's a standard approach in industry. I still believe it is more common to fully leverage the document (i.e., 100-word passage) rather than further chunking it into chunk_size=64. However, I think a further revision of writing should be enough.
> >
> > For W2, I appreciate the purpose of ablation study in isolating the effect of each term. While empirical analysis could be performed, I think the left term does influence right term, making the results not entirely interpretable.
> >
> > For W3, DRAGON evaluates on the full BeIR and ColBERTv2 does on 13 datasets (BeIR-13). (For ColBERT, do you mean v2 https://arxiv.org/pdf/2112.01488?). When studying BeIR, I believe it is necessary to either provide the full results or provide a justification beyond random selection. Also, I was wondering why nDCG@10, which is the most common metric for BeIR, was not reported.

---

> ### Author Response · Authors · 2024-11-27
>
> Thanks a lot for your questions.
>
> > Thank you for your reply, which has addressed W1, Q1, Q2. Regarding Q3, I appreciate it's a standard approach in industry. I still believe it is more common to fully leverage the document (i.e., 100-word passage) rather than further chunking it into chunk_size=64. However, I think a further revision of writing should be enough.
>
> Chunk size is a hyperparameter that we determined using our dev set and then fixed for evaluation across all sets in this paper. In practical scenarios, a single document may contain thousands of tokens. Creating a single embedding vector for such lengthy texts can lead to information loss; therefore, it is beneficial to split them into smaller chunks. While setting chunk_size=128, (equivalent to approximately 100-word passages), can help reduce the indexing cost of the retriever, the optimal selection of chunk_size is a topic that can be explored independently of this work. The primary objective of our paper is to introduce a novel framework that enhances the recall performance of retrievers without requiring additional training, and we leave the investigation of chunk size optimization for future studies.
>
> &emsp;
>
> > For W2, I appreciate the purpose of ablation study in isolating the effect of each term. While empirical analysis could be performed, I think the left term does influence right term, making the results not entirely interpretable.
>
> Thank you for your comment. Our approach is to add the doc level embedding (the right term) on top of the current standard dense retrieval approach, i.e. the chunk level embedding (the left term). Therefore we keep the left term active while adjusting the other fields (title, query) to see their impacts on the final results. To minimize confusion, we have removed the row row corresponding to the case where $w_c=1.0$ in our ablation study and only focus on discussing the impact of synthetic query and title fields.
>
> &emsp;
>
> > For W3, DRAGON evaluates on the full BeIR and ColBERTv2 does on 13 datasets (BeIR-13). (For ColBERT, do you mean v2 https://arxiv.org/pdf/2112.01488?). When studying BeIR, I believe it is necessary to either provide the full results or provide a justification beyond random selection. Also, I was wondering why nDCG@10, which is the most common metric for BeIR, was not reported.
>
> Yes, we mean the ColBERTv2. For BEIR, we initially selected a subset of datasets with varying number of documents to ensure representativeness. However, I concur that evaluating on the complete BEIR datasets would be beneficial. Given the limited time and compute resources we have now, we will include these evaluations here later and update them in the final version of the paper.
>
> For nDCG@10, we reference the Contriever [1] paper, which states: "While nDCG@10 is the main metric of BEIR, we are more interested in the Recall@100 to evaluate bi-encoders, as our goal is to develop retrievers that can be used in ML systems. Moreover, in many settings, retrieved documents can be re-ranked with a more powerful model such as a cross-encoder, thus improving the nDCG@10."
>
> Similarly, our proposed method was originally designed for the information retrieval component of a Retrieval-Augmented Generation (RAG) system [2]. Typically, we retrieve 3 to 10 documents for each response generation of LLM. The LLM is then better equipped to relate the user query to the most relevant content among the retrieved documents to generate final answers. Therefore, we are more focused on R@3 and R@10. We have also included the nDCG@10 scores on several BEIR subsets for our method below for reference.
>
> | Model      | Metrics | ArguAna | FIQA   | Quora  | SciDocs | SciFact |
> |------------|---------|---------|--------|--------|---------|---------|
> | Contriever*| R@3     | 0.2468  | 0.3690 | 0.8488 | 0.2440  | 0.5996  |
> |            | R@10    | 0.5825  | 0.5174 | 0.9434 | 0.4030  | 0.7259  |
> |            | nDCG@10 | 0.2691  | 0.3604 | 0.8131 | 0.2460  | 0.5790  |
> | Dragon*    | R@3     | 0.4196  | 0.3950 | 0.9098 | 0.2770  | 0.6393  |
> |            | R@10    | 0.7482  | 0.5353 | 0.9698 | 0.4550  | 0.7638  |
> |            | nDCG@10 | 0.3678  | 0.3853 | 0.8726 | 0.2825  | 0.6290  |
>
>
> [1] Izacard, Gautier, Mathilde Caron, Lucas Hosseini, Sebastian Riedel, Piotr Bojanowski, Armand Joulin, and Edouard Grave. "Unsupervised dense information retrieval with contrastive learning." arXiv preprint arXiv:2112.09118 (2021).
>
> [2] Lewis, Patrick, Ethan Perez, Aleksandra Piktus, Fabio Petroni, Vladimir Karpukhin, Naman Goyal, Heinrich Küttler et al. "Retrieval-augmented generation for knowledge-intensive nlp tasks." Advances in Neural Information Processing Systems 33 (2020): 9459-9474.
>
> &emsp;
>
> Furthermore, if our actions and responses have effectively addressed your questions and concerns, we would greatly appreciate it if you could consider increasing your rating of our paper.  Your support would be immensely appreciated!

---

> > ### Author Response · Authors · 2024-12-04
> >
> > Please refer to the table in the main section for results on BEIR and nDCG@10.

---

### Author Response · Authors · 2024-11-22
**Rebuttal**

# Rebuttal

We sincerely appreciate the insightful feedback provided by all reviewers. We are pleased to note that the reviewers recognize our paper as achieving robust, significant, and substantial results [**nf7F, z7Jy, fyxe, ZRAX**] through a straightforward method [**z7Jy, ZRAX**] that is easy to implement.


## Highlights and Common Questions
We summarize several highlights and common feedback as below.


### Our novelty is primarily derived from the following aspects
1. We present a **training-free**, straightforward yet effective method that demonstrates improvements over existing state-of-the-art (SoTA) retrievers [**nf7F, z7Jy, fyxe, ZRAX**].
2. Our approach maintains inference speed without any sacrifice, and the total cost is minimized as the retriever index is pre-computed only once prior to the inference step [**nf7F**]. Additionally, the index size remains consistent with that of single-vector dense retrieval while including richer information from multiple fields [**z7Jy**].
3. We provide valuable insights, such as the impact of titles on information retrieval and the corresponding methods to employ when titles are absent [**ZRAX**].
4. Our solution is **model-agnostic** (also multilingual-agnostic) and can be applied to both Bi-encoder and token-level late-interaction models [**nf7F**].


### We have enhanced the presentation of the paper with the following improvements
1. We have added a quantitative summary of the augmentation details, including the number of queries/tokens generated [**nf7F, ZRAX**] and the query match ratio [**ZRAX**], in Table 4, Section 4.5 with **blue** color, where we also discuss the cost and latency analysis of our proposed method. In short, the cost impact is limited.
2. We have included qualitative examples [**nf7F, z7Jy, ZRAX**] of synthetic queries: Table 11 in the Appendix, highlighted in **blue** color, uses four documents as examples to illustrate the sample synthetic relevant queries generated by LLMs.
3. We have improved the description of hyperparameter selection and experiments details [**nf7F, z7Jy**], highlighted in **orange** color in Section 4.3.
4. We also improved our notations in equations in **teal** color to address reviewer’s questions [**z7Jy**] in Section 3.2.1 regarding how synthetic queries work in doc-level embedding.

---

> ### Author Response · Authors · 2024-12-04
> **Follow up on full BEIR Experiment**
>
> As requested by reviewer [**nf7F, ZRAX**], we assessed on the whole BEIR set using Contriever as the base model and report NDCG@10 along with R@3 an R@10. We can see the metric improvement by our method is substantial and consistent acoss each dataset.
>
> |  | Contriever  | | | Contriever* | (our method)             | |
> |---------------|:----------------|:--------|:---------|:--------|:--------|:---------|
> |               | R@3                      | R@10   | NDCG@10 | R@3    | R@10   | NDCG@10 |
> | ArguAna       | 0.3030                   | 0.5498 | 0.3317  | **0.3172** | **0.6095** | **0.3481**  |
> | FiQA          | 0.1895                   | 0.2993 |  -    | **0.3690**  | **0.5174** | **0.2866** |
> | Quora         | 0.8653                   | 0.9464 | 0.8311  | **0.8687** | **0.9517** | **0.8332** |
> | Scidocs       | 0.1560                   | 0.2930 | 0.1700  | **0.2430** | **0.4040** | **0.2460**  |
> | Scifact       | 0.5410                   | 0.6934 | 0.5310  | **0.6005** | **0.7259** | **0.5796**  |
> | Climate-FEVER | 0.0612                   | 0.1199 | 0.0725  | **0.2593** | **0.4541** | **0.2784**  |
> | MS MARCO      | 0.6744                   | 0.7907 | 0.6314  | **0.7907** | **0.8837** | **0.7210**  |
> | DBPedia       | 0.4825                   | 0.6425 | 0.4498  | **0.6125** | **0.7750** | **0.5774**  |
> | Touche-2020   | 0.5918                   | 0.6939 | 0.5110  | **0.8163** | **0.8776** | **0.6987**  |
> | NFCorpus      | 0.3065                   | 0.5139 | 0.3233  | **0.5263** | **0.6409** | **0.4808**  |
> | Trec-COVID    | 0.5200                   | 0.7600 | 0.5083  | **0.8800** | **0.9600** | **0.8294**  |
> | CQADupStack   | 0.1639                   | 0.2402 | 0.1643  | **0.2948** | **0.6412** | **0.2940**  |

---

### Meta-Review · Area_Chair_M5XY · 2024-12-18

**Metareview:**

This paper proposes an LLM-based method for document expansion in document retrieval. It enhances each document with synthetic queries, titles, and chunks generated by the LLM. The proposed method is training-free and model-agnostic because of the use of LLM. The method demonstrates substantial performance improvements across various retrievers (Contriever, DRAGON, ColBERTv2) and benchmarks (LoTTE, BEIR). Reviewers found the approach straightforward (z7Jy, ZRAX) and practical to implement (nf7F, z7Jy), with substantial empirical gains (fyxe, ZRAX).

The reviewers also raised several concerns:
1. The high inference cost of LLM-generated augmentations and the unfair comparison with previous methods in terms of inference efficiency (z7Jy, ZRAX).
2. The limited novelty, as the method is incremental with respect to existing document augmentation techniques (z7Jy, fyxe).
3. Concerns about the evaluation metrics (nf7F, z7Jy) and datasets (nf7F, ZRAX).

During the rebuttal, the authors provided additional results to address concern 3; however, concerns 1 and 2 were not sufficiently addressed. A rejection is recommended. We encourage the authors to improve the current version based on reviewers’ feedback for future publication.

**Additional Comments On Reviewer Discussion:**

Reviewers also raised several concerns on high inference cost, limited novelty, and evaluation. During the rebuttal the authors provided additional results to address concern on evaluation; however, concerns on cost and novelty were not sufficiently addressed.

---

### Decision · Program_Chairs · 2025-01-22

Reject